# An atypical F-type ATPase is necessary for the function of the antibody cleavage system MIB-MIP in mycoplasmas

Julien Berlureau[1], Robin Anger[1], Emilie Beaulieu[1], Geraldine Gourgues[1], Laure Bataille[1], Jade Jaubert[2], Carole Lartigue[1], Pascal Sirand-Pugnet[1], Yonathan Arfi[1]*

**1** University Bordeaux, INRAE, UMR BFP, Villenave d'Ornon, France, **2** University Bordeaux, Bordeaux Proteome, Bordeaux, France

* yonathan.arfi@u-bordeaux.fr

## Abstract

Mycoplasmas are genome-reduced, parasitic bacteria colonizing a broad range of organisms, including most livestock species and humans. They are often pathogenic, and have evolved an arsenal of mechanisms to counter their host's immune response. Among these, the Mycoplasma Immunoglobulin Binding/Protease (MIB-MIP) system appears to be particularly important, and is conserved in the majority of mycoplasma species. Through genomics analysis, we show that MIB-MIP systematically co-occurs with a cluster of 7 coding sequences corresponding to an atypical Type 3 F-ATPase termed "$F_1$-like $X_0$". Working in the model organism *Mycoplasma mycoides* subsp. *capri*, we first performed a proteomics analysis to confirm that this ATPase is indeed expressed. We then generated two mutant strains in which the putative ATPase was either fully deleted, or rendered catalytically inactive through replacement of a conserved lysine residue in the Walker A motif of the ATPase β-like subunit. Functional assays in presence of immune serum showed that both mutants are unable to protect themselves from agglutination by immunoglobulins despite the MIB-MIP system still being present. We then attempted to affinity-purify the atypical $F_1$-like $X_0$ ATPase from the membrane of its native host. Although the complex appears to be labile, under cross-linking conditions we were able to co-purify all its predicted components, as well as both MIB and MIP. These results allow us to attribute a function to Type 3 F-ATPase, namely to participate in the evasion from the humoral immune response in mycoplasmas, in conjunction with the MIB-MIP system through a currently unknown mechanism.

## Author summary

Mycoplasmas are small parasitic bacteria responsible for infections in a range of mammalian hosts, including humans and most livestock species. They take part in a constant arms race, evolving new mechanisms to try to evade or defeat their

**Data availability statement:** All relevant data are within the manuscript and its Supporting Information files, with the exception of raw sequencing reads which have been deposited in the SRA under Accession Number PRJNA1359759.

**Funding:** This work was financed by the French National Agency for Research (ANR) grant ANR-21-CE44-0002 ENIgMA (Yonathan Arfi). The funders had no role in the study design, data collection and analysis, the decision to publish, or the preparation of the manuscript.

**Competing interests:** The authors have declared that no competing interests exist.

hosts' complex immune systems. In this study, we describe an atypical F-ATPase known as Type 3, or $F_1$-like $X_0$, and show that it is genetically linked to MIB-MIP, as system dedicated to the capture and destruction of antibodies. Through phenotypic comparison of several mutant mycoplasma and co-purification attempts, we then show that this F-ATPase is indeed functionally linked to MIB-MIP, and even essential for the system to function. However, at this point the exact mechanism of interaction between all the partners is still unknown. Our study highlights the evolution of a complex molecular machine in mycoplasmas to combat the humoral immune response of their hosts. As the MIB-MIP- $F_1$-like $X_0$ ATPase system is conserved in mycoplasmas, it appears to be a promising target for the development of specific inhibitors, which could act as complement to classical antibiotics.

## 1. Introduction

The term "mycoplasmas" refers to a polyphyletic set of bacteria belonging to the Class *Mollicutes*, which are characterized by their small genomes (0.6-1.4 Mb) resulting from a fast and reductive evolution from a common ancestor with *Firmicutes* [1]. They are obligate parasites and rely on their hosts to supply them with an array of essential components, as these bacteria lack a number of important metabolic pathways including nucleotides and lipids synthesis [2–4].

Mycoplasmas can be found in a large array of vertebrate hosts, including humans and most livestock species, and they are causes of concerns in both the veterinary and human medicine fields [5–9]. Indeed, although some mycoplasmas are commensal, most are pathogenic and colonization is often associated with high morbidity and low mortality inflammatory diseases such as walking pneumoniae, arthritis or urethritis. These infections are often chronic and mycoplasmas are able to persist in the host despite an apparently normal immune response, with both the humoral and cell-mediated processes involved [10–14]. The ability of mycoplasmas to evade immunity appears to involve a large set of strategies that vary depending on the species, including (but not limited to) intracellular colonization, surface protein expression switching, trafficking of cytoplasmic proteins at the cell surface, and high frequency erroneous translation [15]. In addition, mycoplasmas also deploy a set of active effectors, predominantly of enzymatic nature, to combat immune factors such as nucleases to degrade Neutrophil Extracellular Traps (NETs) and immunoglobulin-specific proteases to cleave antibodies in the hinge region [16–18].

Another type of active system targeting antibodies has recently been described and is specific to mycoplasmas: the Immunoglobulin-Blocking Proteins (IBPs). IBPs are comprised of two related subsets of effectors that can bind to the Light Chain of immunoglobulins. The first set corresponds to Protein M homologs, which are found in a small group of mycoplasmas, including the human pathogens *Mycoplasma genitalium* and *Mycoplasma pneumoniae* [19,20]. These effectors, located at the cell surface, can block the interaction between an antibody and its cognate antigen by

capping the paratope with their C-terminal domains. However, Protein M cannot interact with antibodies that are already bound to the antigen, due to steric hindrance. The physiological relevance of Protein M currently remains unelucidated.

The second set of IBPs corresponds to homologs of the Mycoplasma Immunoglobulin Binding protein [21,22]. MIB is structurally related to Protein M, but differs in its functionality. Indeed, MIB has the ability to bind to antibodies that are already bound to their antigens, and to promote the dissociation of these immune complexes by twisting the variable domains of the Heavy and Light Chains of immunoglobulins. Interestingly, MIB has been shown to work in combination with the serine protease Mycoplasma Immunoglobulin Protease, forming the MIB-MIP system. After capturing an antibody, MIB can recruit MIP which in turn will interact with the Heavy chain of the antibody and cleave it proteolytically between the $C_H 1$ and $V_H$ domains. Broken antibodies are released in the environment and are no longer able to bind to their corresponding antigens (Fig 1A). The MIB-MIP system has a much broader distribution than Protein M, as it is found in the vast majority of mycoplasmas and has been shared through horizontal gene transfer between species of the *Hominis* group and the species of the *Spiroplasma* and *Pneumoniae* groups [22–25]. Most species encode more than one set of MIB-MIP coding sequences, with up to four paralogs of each, and sometimes miss-matched numbers of MIBs and MIPs. Owing to its function and widespread conservation, the MIB-MIP system appears to be a crucial element of the immune evasion by mycoplasmas.

Intriguingly, *in vitro* biochemical and structural characterization experiments have shown that MIB and MIP form a highly stable ternary complex with the antibody, even after cleavage has occurred [21,22]. This observation is in stark contrast with *in cellulo* and *in vivo* experiments in which the antibodies that have been processed by MIB-MIP are found free in the medium [21,23,25–27]. This discrepancy suggests that an unknown release factor might be involved to reset the MIB-MIP system in the cells.

In our model system *Mycoplasma mycoides* subsp. *capri* (*Mmc*), the locus encoding the four MIB-MIP paralogs is part of a larger region that also contains a cluster of seven genes. This cluster has first been described in the closely related *Mycoplasma mycoides* susbp. *mycoides* (*Mmm*) and putatively encodes an atypical Type 3 ATPase termed "$F_1$-like $X_0$" [28]. This ATPase is specific to mycoplasmas, and is predicted to contain homologs of the alpha (α), beta (β), gamma (γ) and epsilon (ε) subunits forming the $F_1$-like domain; as well as three proteins with no known homologs: a small cytoplasmic protein and two membrane proteins with respectively twelve and two transmembrane segments forming the $X_0$ domain. The topology, stoichiometry and function of this putative $F_1$-like $X_0$ ATPase complex are currently unknown, and only ATP hydrolysis activity and non-essentiality have been demonstrated [28]. Two other types of F-ATPase are found in mycoplasmas: the ubiquitous Type 1 which corresponds to the essential $F_1 F_0$ ATP synthase; and the rare Type 2 which corresponds to the G1-ATPase, the molecular motor involved in energizing the gliding motility in *Mycoplasma mobile* [29,30].

In this study, we first analyzed the $F_1$-like $X_0$ ATPase of *Mmc* using structure prediction tools, and validated its expression through proteomics. We then showed through genomics analysis that the genes encoding MIB-MIP systematically co-occur with the cluster encoding the $F_1$-like $X_0$ ATPase. Based on this strong genetic linkage, we then investigated the role of the ATPase in the cleavage of antibodies by MIB-MIP. To do so, we generated a pair of mutants in *Mmc*: a ΔATPase strain in which the 7 genes cluster is deleted; and a β-like subunit mutant in which the conserved lysine in the Walker A motif is substituted with an alanine. Both mutants were then tested for their ability to evade agglutination by immune serum and to cleave immunoglobulins. We then attempted to isolate the $F_1$-like $X_0$ ATPase complex and assessed the co-purification of MIB and MIP with the $X_0$ domain.

## 2. Results

### 2.1. Description and structure prediction of the putative $F_1$-like $X_0$ ATPase of *Mmc*

The MIB-MIP system was first identified and characterized in the model species *Mycoplasma mycoides* subsp. *capri* strain GM12. In this organism, four contiguous pairs of MIB-MIP coding sequences are found in the genome, with the

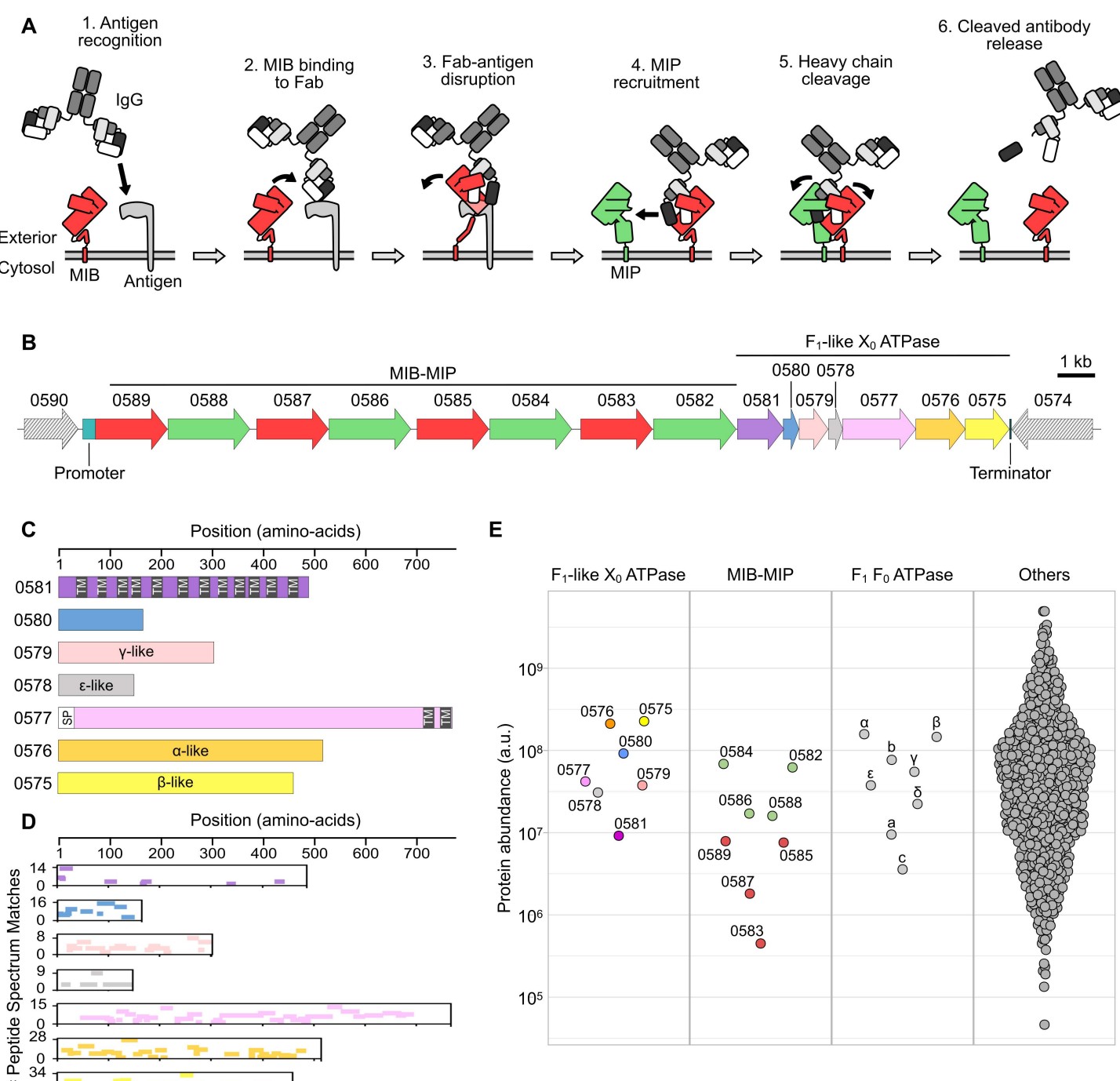

**Fig 1. The F₁-like X₀ ATPase locus is expressed in *Mmc*. A)** Schematic view of the current model of the MIB-MIP system in mycoplasmas, and its role in evading the humoral components of the immune response. **B)** Schematic representation of the genomic locus encoding MIB-MIP and the $F_1$-like $X_0$ ATPase in *Mmc* GM12. MIBs are colored in red, MIPs in green, and the ATPase proteins in purple, blue, salmon, grey, pink, orange and yellow. This color code is used throughout the Figs. The name of each locus is based on the last four characters of the corresponding mnemonics (e.g., 0589 corresponds to MMCAP2_0589). This naming scheme is used throughout the Fig. Hatched arrows correspond to the two loci flanking the region of interest. The position of the predicted promoter and terminator is indicated. **C)** Schematic representation of the seven proteins encoded by the $F_1$-like $X_0$ ATPase locus. For each protein, the size in amino acids is represented, as well as the presence of predicted transmembrane domains (black box labeled "TM") and of a predicted signal peptide (white box labeled "SP"). **D)** Mapping of the peptides detected in the proteomics dataset for *Mmc* GM12 for the seven proteins of the $F_1$-like $X_0$ ATPase. For each protein, the position of a unique peptide is denoted by a colored bar. The number of Peptide Spectrum

Matches for each mapped peptide is denoted by the vertical position of the colored bars. **E)** Dot-plot representation of the abundance (arbitrary unit) of each protein in the proteomics dataset for *Mmc* GM12. The values are the mean calculated from three biological replicates. For the $F_1F_0$ ATPase, the names of the subunits are given instead of the shortened mnemonics.

mnemonics MMCAP2_0589-MMCAP2_0582. Immediately downstream, and seemingly in operon, a cluster of seven coding sequences corresponding to the mnemonics MMCAP2_0581-MMCAP2_0575 encodes a putative $F_1$-like $X_0$ ATPase [28] (Fig 1B). The characteristics of the seven predicted proteins match those previously identified in *Mycoplasma mycoides* subsp. *mycoides*. MMCAP2_0581 encodes a 57 kDa protein with no known homolog and twelve predicted transmembrane segments. MMCAP2_0580 encodes a 19 kDa protein, predicted to be cytoplasmic and with no known homolog. MMCAP2_0579 corresponds to a 36 kDa protein with a strong homology to the γ subunit of $F_1F_0$ ATP synthase which normally forms the central shaft of the $F_1$ domain. MMCAP2_0578 encodes a 17 kDa protein homologous to the ε subunit of F-type ATP synthase, which normally acts as a catalytic regulator of the $F_1$ domain. MMCAP2_0577 is predicted to be an 87 kDa membrane protein, with a N-terminal signal peptide and two C-terminal transmembrane domains. Finally, MMCAP2_0576 and MMCAP2_0575 encode two proteins of 58 kDa and 51 kDa respectively, predicted to be homologous to the α and β subunits of the $F_1F_0$ ATP synthase (Fig 1C).

Alphafold3 was used to predict the structures of the seven individual monomers, as well as various combinations of multimers to predict potential complexes (S1 Fig). High confidence predictions were obtained for the α-like, β-like, γ-like and ε-like subunits of the $F_1$-like $X_0$ (MMCAP2_0579 pTM = 0.82; MMCAP2_0578 pTM = 0.89; MMCAP2_0576 pTM = 0.90; MMCAP2_0575 pTM = 0.93) (S1A Fig). Alignment to the experimentally resolved structure of the *E. coli* $F_1F_0$ ATPase showed that they all closely resemble their $F_1$ counterparts. The main difference is predicted for MMCAP2_0576, with a clear lack of the short N-terminal helix found in the $F_1$ α-subunit, which normally interacts with the δ and b-b' subunits and stabilize the stator and peripheral stalk [31]. Multimer predictions performed using a stoichiometry 3α:3β-like:1ε-like:1γ-like yielded a predicted $F_1$-like complex, with high confidence score (iPTM = 0.75; pTM 0.77) (S1B Fig). In this predicted complex, the α-like and β-like subunits form a ring in which the γ-like coiled-coil is wedged, with ε-like subunit located next to its base. Meanwhile, the predictions for the components of the putative $X_0$ were less confident (MMCAP2_0581 pTM = 0.87; MMCAP2_0580 pTM = 0.85; MMCAP2_0577 pTM = 0.56) although they matched previous inferences for the two putative membrane proteins: MMCAP2_0581 is predicted to form a bundle of 12 transmembrane alpha-helices, and the model of MMCAP2_0577 shows a long N-terminal disordered region followed by two globular domains connected by a linker and with two transmembrane helices in C-terminal. Finally, MMCAP2_0580 was predicted to form a globular protein which could interact with the C-terminal of the α-like subunit (S1C-S1D Fig). Only low-confidence models of complexes were obtained when the seven proteins were tested together at various stoichiometries (iPTM = 0.66; pTM 0.69) (S1D Fig). In these predictions, the $F_1$-like domain was predicted to interact directly with MMCAP2_0581 *via* the ε-like and γ-like proteins, and MMCAP2_0577 was systematically placed in a distal position on the other side of MMCAP2_0581.

### 2.2. The $F_1$-like $X_0$ ATPase is expressed in *Mmc*

In order to confirm the expression of the putative $F_1$-like $X_0$ ATPase in our model organism, liquid-chromatography with tandem mass-spectrometry was used to analyze the proteome of *Mmc* GM12. Whole cell extracts were prepared from cells grown in axenic conditions, and collected in late log-phase. Peptides were confidently detected for all seven predicted proteins, thus confirming the expression of this $F_1$-like $X_0$ ATPase (Fig 1D and S1 Table). High peptide counts and coverage were obtained for all the soluble proteins and for the soluble domains of the two membrane proteins, but coverage was lacking for the transmembrane domains (Fig 1D) as expected [32]. The calculated abundances for the $F_1$-like $X_0$ ATPase proteins ranged between 9.2E + 06 to 2.3E + 08, and correspond to a medium expression level when compared to the rest of the proteome abundances (median = 3.8E + 07; Q1 = 9.7E + 06; Q3 = 1.2E + 08) (Fig 1E). These values are

similar to those obtained for the eight proteins forming the $F_1F_0$ ATP synthase, with matching lower abundances for the membrane proteins and higher abundances for the cytoplasmic proteins. Slightly lower abundances were observed for the four MIPs (1.6E+07 to -6.9E+07), while these values were an order of magnitude lower for the MIBs (4.5E+05 to 7.9E+06). These relative differences between MIBs and MIPs abundances have been observed in earlier studies, however it is still unclear whether the low amount of detected MIBs is biologically relevant or artefactual [21].

## 2.3. The $F_1$-like $X_0$ ATPase cluster is widespread in mycoplasmas and systematically associated with MIB-MIP

Given the genetic proximity in *Mmc* between the loci encoding the MIB-MIP paralogs and the $F_1$-like $X_0$ ATPase, we analyzed the potential co-occurrence of these two systems in *Mollicutes* genomes. Based on sequences available in the Molligen 4.0 database (https://services.cbib.u-bordeaux.fr/molligen4), we identified putative MIB and MIP homologs in 126 strains from 56 species of *Mollicutes* (Fig 2 and S2 Table). In the totality of these strains (126/126), we were able to identify a predicted homolog for each of the seven proteins of the $F_1$-like $X_0$ ATPase. In 89% of the species (50/56) and 94% of the strains (119/126) the seven coding sequences were clustered together, and with a conservation of synteny when compared to *Mmc*. Among the exceptions, *Mycoplasma alkalescens*, *Mycoplasma canadense*, *Mycoplasma arginini* and *Mycoplasma auris* shared a common organization with the α-like and β-like coding sequences clustered together on a distinct locus. Given the close phylogenetic relationship between these three species, the scattering of the cluster probably occurred in a common ancestor. Meanwhile, in *Mycoplasma arthritidis* two $F_1$-like $X_0$ ATPase clusters are found, one of which presents an inversion of the α-like and β-like loci.

In addition to its systematic co-occurrence with MIB-MIP, we also observed that in 85% of the species (48/56) and 90% of the strains (114/126) the $F_1$-like $X_0$ ATPase cluster is located immediately next to a locus encoding either a MIB or MIP homolog. However, the global organization of these genes in a single operon as seen in *Mmc* is not as widespread, with only 45% of the species (25/56) and 60% of the strains (76/126) exhibiting the same genetic organization as in *Mmc*. A scattering of MIB and/or MIP coding sequences away from the $F_1$-like $X_0$ ATPase cluster is often observed, in particular in species from the Hominis group. Phylogenetic analysis of the predicted homologs of the seven proteins of the $F_1$-like $X_0$ ATPase (S2-S8 Figs) suggests that this cluster was transferred horizontally several times during the *Mollicutes*' evolution. The first case is seen between the phylogenetically distant species of the *Mycoplasma bovis-Mycoplasma agalactiae* group and the Mycoides cluster which have closely related $F_1$-like $X_0$ ATPase. The second occurrence is between the phylogenetically distant *Mycoplasma hominis* and *Ureaplasma spp*. Such a Horizontal Gene Transfer event could also explain why the MIB -MIP - $F_1$-like $X_0$ ATPase operonic structure appears to be maintained in these species, while being lost in their closest relative species.

Taken together, these data indicate a strong genetic linkage between the MIB-MIP system and the $F_1$-like $X_0$ ATPase, in turn suggesting that a functional link may exist as well.

## 2.4. Generation of *Mmc* mutant strains impaired for the $F_1$-like $X_0$ ATPase

In order to probe this potential functional linkage, we have generated three mutants in *Mmc* 1.1 [33]. This strain is a genetically modified derivative of *Mmc* GM12, whose genome was marked using a transposon carrying a yeast centromeric plasmid and subsequently cloned in yeast. This process enables us to perform advanced genetic editing of the genome, in yeast, before performing genome transplantation to generate the corresponding mutant strain [26,33,34]. In the first mutant, termed "*Mmc* ΔATPase", we have performed the clean deletion of the locus containing the seven coding sequences of the $F_1$-like $X_0$ cluster, from the ATG initiation codon of MMCAP2_0581 to the TAA Stop codon of MMCPA2_0575. This deletion maintains the MIB-MIP encoding locus upstream intact, and juxtaposes the putative operon terminator downstream of the intergenic spacer following MMCAP2_0582 (Fig 3A).

The second mutant was designed to probe the reliance of the $F_1$-like $X_0$ ATPase on ATP. Indeed, sequence alignments of the α-like and β-like subunits homologs (MMCAP2_0576 and MMCAP2_0575 respectively) with their $F_1F_0$ counterparts

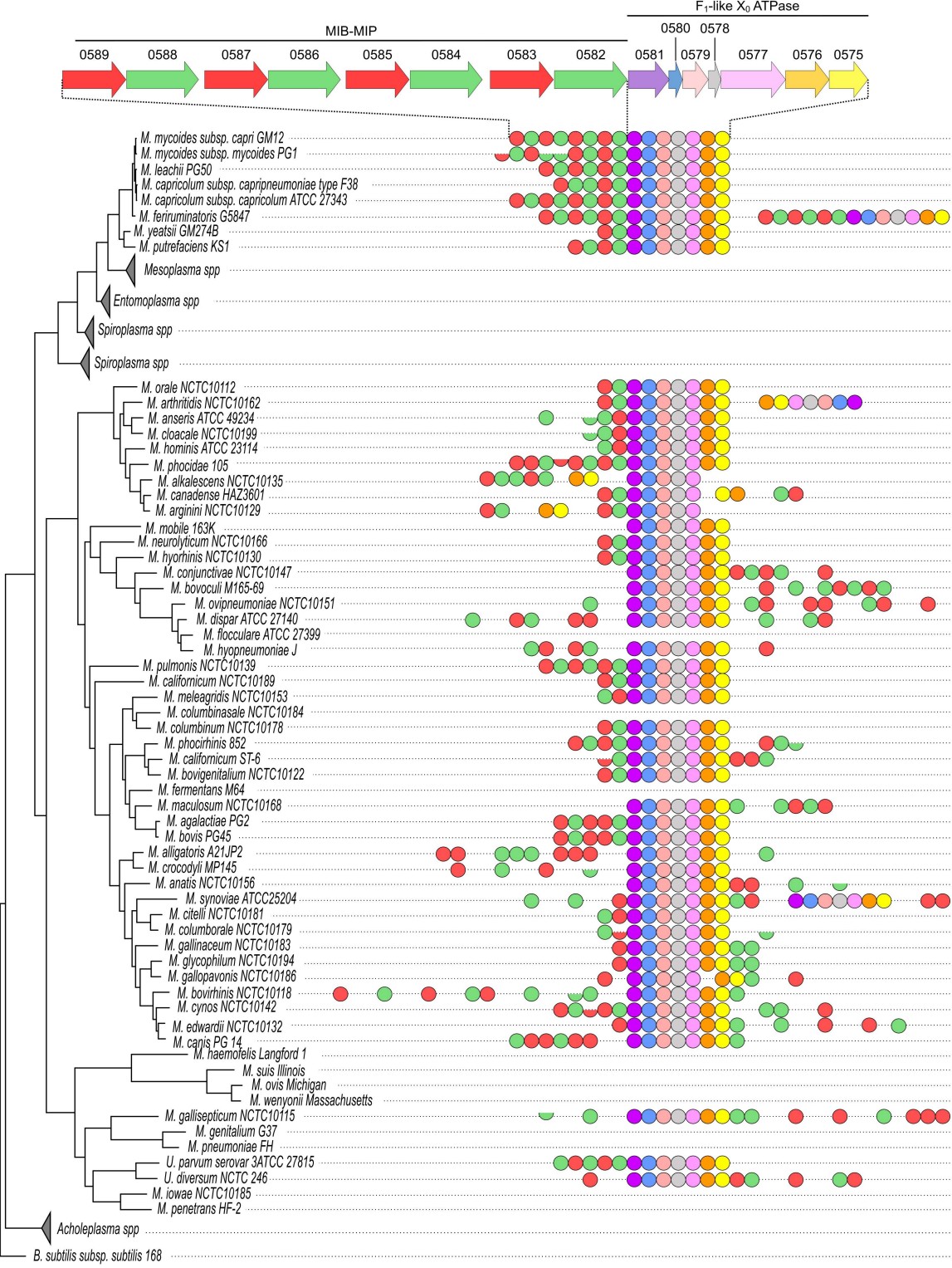

**Fig 2. Occurrence of MIB, MIP and F₁-like X₀ ATPase subunit homologs in *Mollicutes*.** A phylogenetic tree was established for a set of 104 representative *Mollicutes* species. The presence in their genome of predicted MIB, MIP or F₁-like X₀ ATPase homologs is depicted by a colored symbol, using the same color code as in Fig 1. The structure of the locus in *Mmc* is shown at the top for reference. A full circle corresponds to homologs with sizes

closely matching that of their *Mmc* counterparts. Half circles correspond to predicted homologs that present an important size reduction (at least 25%) compared to the reference proteins from *Mmc*, indicative of a potential truncation of the coding sequence. Adjacent symbols indicate homologs that are found at adjacent loci. Symbols that are separated (non-touching) indicate homologs that are found at different loci. The distance between the symbols is arbitrary and does not correspond to a distance between the loci on the genome. Branches corresponding to *Mesoplasma spp.*, *Entomoplasma spp.*, *Spiroplasma spp.* and *Acholeplasma spp.* did not contain any predicted homolog and are collapsed.

revealed a strong conservation of the Walker A motif GXXXXGKT [35,36] in both proteins (S9-10 Figs). We elected to target the highly conserved lysine residue of the Walker A motif of the β subunit (Fig 3A), which normally coordinates the β- and γ-phosphates of ATP, positioning it for hydrolytic attack. However, to do so required the use of a CRISPR-Cas9 genome editing tool and thus also warranted that we edited out the Protospacer Adjacent Motif (PAM) associated with the guide RNA target sequence. However, synonymous recoding of the PAM was not possible, and it was therefore necessary to perform a non-synonymous recoding which yielded the substitutions G3S and K4R. In order to confirm that these two undesired modifications in the N-terminal part of MMCAP2_0575 had no effect, we have thus generated an independent mutant, termed "*Mmc* 0575-recode" in which the substitutions G3S and K4R are present, but the lysine 152 is maintained (Fig 3A). In parallel, we have also generated the strain termed "*Mmc* 0575-K152A" in which the lysine 152 of MMCAP2_0575 was substituted by an alanine, and which also carries the G3S and K4R substitutions (Fig 3A).

After molecular cloning of the different cassettes and plasmids necessary, editing of the *Mmc* 1.1 genome carried in yeast was successfully performed, and viable transplants were obtained for all the strains (S11-12 Fig). After screening, a single clone for each mutant strain was kept and further analyzed. We first performed whole genome sequencing, using both long and short reads, in order to verify if the genetic sequences of the mutants matched the expected design and to check if undesired genetic mutations had occurred. For all the mutants, the correct sequence was obtained for the MIB-MIP- $F_1$-like $X_0$ ATPase locus, with the expected mutations. Meanwhile, elsewhere in the genome, no major genetic re-arrangement could be observed, and only Single Nucleotides Polymorphisms (SNPs) and short insertion-deletion (INDELs) events had occurred (S3 Table). None of these mutations appeared to be directly correlated to the phenotypes observed in our subsequent experiments, as the majority was found in intergenic regions, in mobile genetic elements (IS1296), or were amino-acids substitutions in proteins of unknown functions. Of note, in the *Mmc* ΔATPase mutant, multiple SNPs and INDELs were found in the gene encoding the 23S rRNA. However, these mutations did not seem to have any significant impact, as the mutant exhibits a seemingly wild-type growth. In addition, the overall number of point mutations was higher in the *Mmc* ΔATPase mutant compared to the two other strains we generated, which could be linked to a higher number of passages.

In order to further validate whether the mutations had any significant undesired impact, we have performed a Western Blot analysis of the expression of the β-like subunit (Fig 3B). To do so, we have used a rabbit anti-sera raised against peptides derived from MSC_0618, the homolog of MMCAP2_0575 in *Mmm* [37]. In the control lane *Mmc* 1.1, the expected band at ~50 kDa was clearly visible. This band was completely absent from the *Mmc* ΔATPase mutant, and present at wild-type levels in both the *Mmc* 0575-recode and *Mmc* 0575-K152A strains. We then proceeded to perform a proteome analysis of the three mutants, in order to assess the expression of the other proteins encoded in the MIB-MIP-ATPase locus (for which no antibody is available), as well as to verify that the correct genomes sequences were matched by the correct corresponding proteomes (Fig 3C and S4 Table). Overall, no major alteration of the proteome was detected, with the majority of the detected proteins presenting a fold-change of their abundances between 0.5 and 2, when compared to the control *Mmc* 1.1 (*Mmc* ΔATPase: median = 0.889, Q1 = 0.7165, Q3 = 1.091; *Mmc* 0575-recode: median = 0.999, Q1 = 0.939, Q3 = 1.065; *Mmc* 0575-K152A: median = 1.095, Q1 = 0.941, Q3 = 1.272). In the specific case of *Mmc* ΔATPase, peptides were confidently detected for the four MIBs and four MIPs, following the same distribution seen in *Mmc* 1.1 (Fig 1D), but no peptide could be confidently detected for the proteins of the $F_1$-like $X_0$ ATPase. Some peptides were nonetheless attributed to MMCAP20576 and MMCAP2_0575, but these correspond to peptides that are shared with the α and β

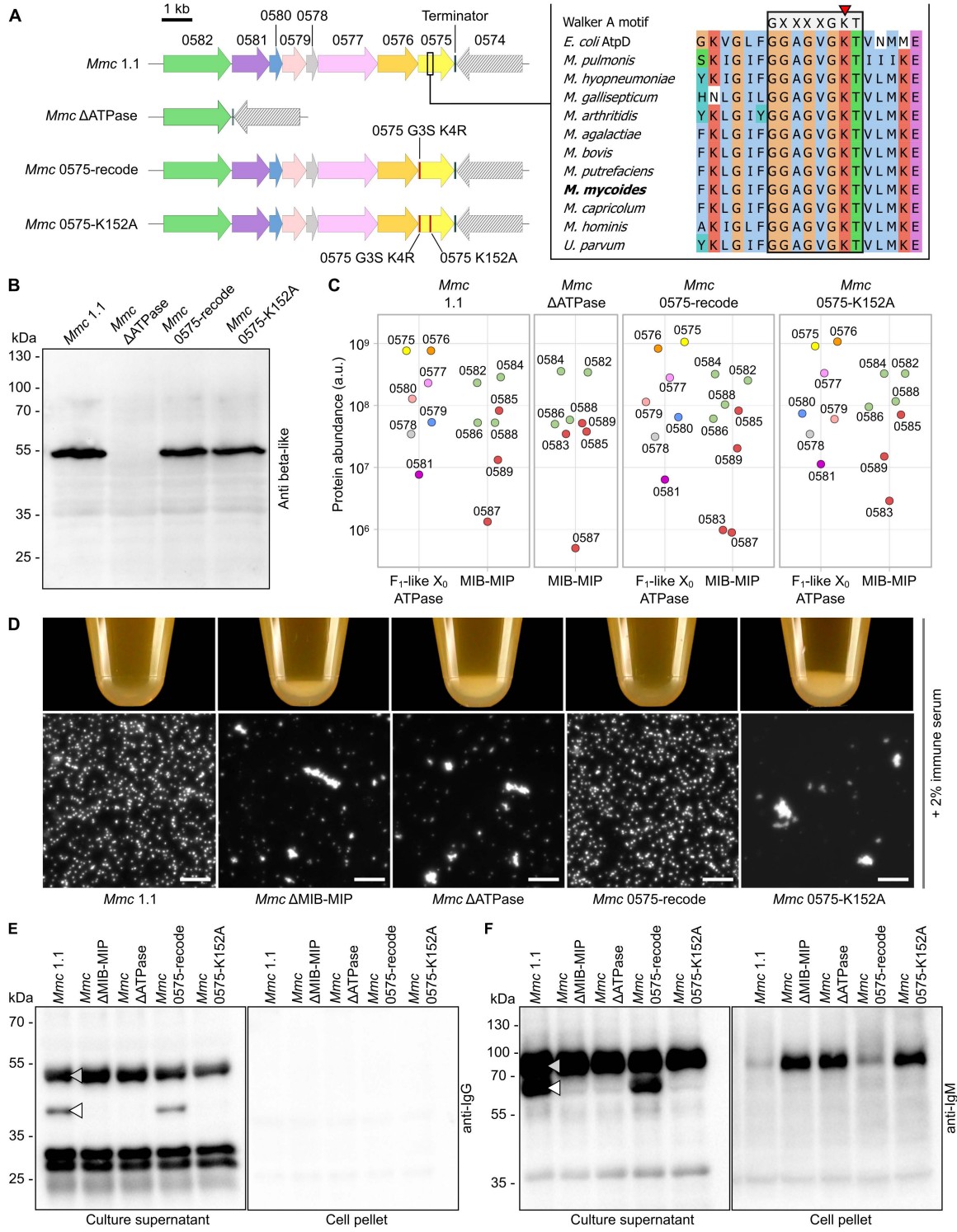

**Fig 3. Validation and phenotypic assessment of *Mmc* mutant strains impaired for the F₁-like X₀ ATPase. A**) Schematic representation of the genomic locus encoding the MIB-MIP-F₁-like X₀ ATPase in the various *Mmc* derivative strains. Only the relevant part of the loci is shown. Inset: the amino-acid sequences of 11 representative β-like homologs were extracted from S2 Table, as well as the sequence of the protein AtpA from *E. coli*, and aligned using ClustalW. The alignment file was visualized using JalView (https://www.jalview.org/), and colored according to the *clustal* palette. The Walker A consensus motif GXXXGKT is noted in grey, aligned with the predicted Walker A motif in the β-like subunits. The conserved Lysine residue is

highlighted by a red arrowhead. **B)** Western blot analysis of the expression of the β-like subunits in the *Mmc* strains. For each *Mmc* derivative, whole cell extracts were generated and separated by SDS-PAGE, then analyzed by Western blotting anti β-like subunit. **C)** Dot-plot representation of the abundance (arbitrary unit) of each protein encoded in the MIB-MIP-$F_1$-like $X_0$ ATPase locus in the proteomics dataset obtained for each *Mmc* strain. For the mutant *Mmc* ΔATPase, no value is reported for the $F_1$-like $X_0$ ATPase subunits as no high confidence peptide could be detected. **D)** Agglutination of cells by immune goat serum. The cells were grown in axenic conditions, in the presence of 2% of serum from a goat experimentally immunized, collected 12 days post-inoculation. The bottom of the micro-centrifuge tubes was photographed (top). After resuspension of the flocculates by inversion of the tubes, a sample was mounted between a glass slide and a coverslip and was imaged on a dark-field microscope (bottom - scale bar: 10 µm. **E-F)** Western blot analysis of the immunoglobulin Heavy Chain cleavage by the MIB-MIP system. Samples corresponding to either culture supernatant or cell pellet derived from the agglutination assays were separated by SDS-PAGE, then analyzed by Western Blotting using primary antibodies targeting either the goat IgG Fc or the goat IgM Heavy Chain. Intact Heavy Chain and MIB-MIP cleaved Heavy Chain are highlighted by a gray and white arrowhead, respectively. *Note: uncropped original images are provided in* S15 Fig.

subunits of the $F_1F_0$ ATPase. Meanwhile, in the cases of both *Mmc* 0575-recode and *Mmc* 0575-K152A, no significant difference to the control could be observed in the abundances of the 13 proteins encoded in the locus. We nonetheless note that our experiments cannot rule out the existence of differences in posttranslational modifications, although these should not affect the MIB-MIP-ATPase components.

Overall, these results indicate that the mutant strains generated are not significantly altered in their global physiology, and that no major alteration of their phenotypes is to be expected (with the exception of the induced mutations).

## 2.5. Functional assessment of the *Mmc* mutant strains

In order to study the involvement of the $F_1$-like $X_0$ ATPase in the antibody cleavage process mediated by MIB-MIP *in cellulo*, we have performed a functional assay based on the incubation of the cells with immune goat sera obtained from an experimentally infected animal. Indeed, we have shown before that the inactivation of MIB-MIP in the mutant *Mmc* ΔMIB-MIP led to the agglutination of the cells by the anti-*Mmc* IgM present in the sera collected 12 days post inoculation [38]. This agglutination was visible to the naked eye, as a white flocculate could be seen at the bottom of the culture, as well as using dark-field microscopy [21]. Here, the control assay with no immune serum showed that all the strains grew normally, with no visible flocculate and well-dispersed cells (S13A Fig). In the presence of immune serum, the control strain *Mmc* 1.1 presented no alteration while the MIB-MIP null mutant *Mmc* ΔMIB-MIP presented the expected agglutinated phenotype (Fig 3D). Interestingly, the complete deletion of the ATPase locus in the *Mmc* ΔATPase strain yielded exactly a very similar agglutinated phenotype, despite the presence of the MIBs and MIPs proteins in wild-type like amounts. Similarly, the mutant *Mmc* 0575-K152A was also agglutinated by the serum, despite the presence of all the proteins encoded in the MIB-MIP-ATPase locus. This phenotype was not linked to the G3S-K4R mutation in MMCAP2_0575, as the strain *Mmc* 0575-recode presented an agglutination-resistant phenotype identical to the wild-type control.

The MIB-MIP induced cleavage of the immunoglobulins Heavy Chain in these assays was then checked by Western Blot, using antibodies against either the goat IgG or goat IgM Heavy Chain, in both the culture supernatant and in the cell pellet. As expected, the negative control condition without serum did not yield any specific signal (S13B Fig). In presence of immune serum, the culture supernatant of the samples from the wild-type *Mmc* 1.1 displayed the expected immunoglobulin cleavage pattern (Fig 3E-F). In the case of IgG (Fig 3E), the intact Heavy Chain (~55 kDa) was accompanied by a specific band at ~44 kDa resulting from the cleavage of the ~11 kDa Variable Heavy domain. In the case of IgM (Fig 3F), the ~79 kDa band for the intact Heavy Chain was accompanied by a ~68 kDa band for the cleaved fragment. In the supernatant of the mutant *Mmc* ΔMIB-MIP, both the 44 kDa and 68 kDa bands were absent, as expected from a MIB-MIP null phenotype. Similarly, for the two mutants impacted in their ATPase (*Mmc* ΔATPase and *Mmc* 0575-K152A) the cleavage of immunoglobulins was also completely abolished. In the cell pellet samples, no signal was obtained for the IgG, as anticipated due to the low amount of anti-*Mmc* IgG in this immune serum. For the IgM, intact antibodies were found only in the cell pellets of the agglutinated mutants: *Mmc* ΔMIB-MIP, *Mmc* ΔATPase and *Mmc* 0575-K152A. Conversely, only a limited signal is visible for the two non-agglutinated strains: the wild-type *Mmc* 1.1 and the wild-type like *Mmc* 0575-recode.

Taken together, these results confirm the functional involvement of the $F_1$-like $X_0$ ATPase in the evasion of the humoral immune response mediated by MIB-MIP.

## 2.6. Purification of the $F_1$-like $X_0$ ATPase subunits

To further study this atypical $F_1$-like $X_0$ ATPase, and gain more insight on its function and the mechanism by which it interacts with MIB and/or MIP, we attempted to purify this complex from its native host. To do so, we elected to add an affinity tag to one of the two membrane proteins. Initial attempts to tag MMCAP2_0581 all failed, as well as early attempts to tag MMCAP2_0577 by positioning a fusion sequence downstream of the signal peptide. We eventually succeeded by adding the sequence of the ALFA-tag, a 13 amino-acids epitope tag for which a cognate Nanobody is available [39], after the residue S96 of MMCAP2_0577 (Fig 4A). This position is downstream of a Glycine-Serine rich region located between residues G48 and S96, which contains 60% of G/S. The addition of the ALFA-tag was performed using the in-yeast genetic editing method mentioned above.

We elected to work in a simplified model by using the mutant strain *Mmc* Rational Operon which was previously generated and validated [21]. This strain has been modified to maintain a single pair of MIB-MIP coding sequence in the locus, by clean deletion of MMCAP2_0589-MMCAP2_0584. This results in a prototypical, rationalized, 1 MIB – 1 MIP – 1 ATPase system (Fig 4A) which is fully functional and exhibit a wild-type like cleavage of immunoglobulins and evasion of serum agglutination. The resulting ALFA-tagged mutant is termed "*Mmc* Rational Operon 0577-ALFA[96]". After molecular cloning of the necessary cassettes and plasmids, editing of the *Mmc* Rational Operon genome was successfully carried-out in yeast, and viable transplants were obtained (S14 Fig). Mutant validation was performed using Western blot to assess the expression of the β-like subunit and ALFA-tag, and the results showed the expected bands at ~50 kDa (β-like) and ~85 kDa (MMCAP2_0577-ALFA[96]) (Fig 4B). An immunoglobulin cleavage assay was also performed to confirm that the addition of the ALFA-tag had not compromised the MIB-MIP-ATPase system's function. Again, the mutant performed similarly to the wild-type, with the presence of a ~44 kDa band in an anti-IgG Western blot on culture supernatant samples (Fig 4B).

An initial fractionation assays was performed, by lysing the cells by sonication and separating the cytoplasmic content from the membranes (Fig 4C). Western blot against the β-like subunit and against the ALFA tag was used to track the location of the putative $X_0$ and $F_1$-like domains. These experiments showed that MMCAP2_0577-ALFA[96] was indeed located in the membrane, but that only a small fraction of the β-like subunits was in the same compartment. In order to purify the putative $X_0$ domain, the washed membrane fraction was collected and solubilized by addition of n-dodecyl-β-D-maltoside (DDM), a detergent that was selected after an initial screening of several detergents (Cube Biotech, Detergent Screening Set Classic and Detergent Screening Set Classic 2, which respectively contain the detergents OG, LDAO, DM, DDM and OTG, and NG, NM, UDM, TDM and CHAPS). The solubilized membrane proteins were subsequently purified by affinity chromatography using a resin functionalized with a high-affinity and high-specificity anti-ALFA tag Nanobody. Analysis of the purified proteins was performed by a combination of stained SDS-PAGE and mass spectrometry (Fig 4D). The results showed the presence of a major band at ~85 kDa, which was confirmed by Western-blot to correspond to an ALFA-tagged protein and identified as MMCAP2_0577. A lighter secondary band, of fainter intensity, is visible at ~40 kDa. This molecular weight does not match any putative component of the $X_0$ domain. Subsequent mass spectrometry analysis of this band indicates with a high confidence that this protein is MMCAP2_0581, as it the most abundant protein in the sample (S5 Table), whose molecular weight is calculated at ~57 kDa based on its amino-acid sequence. It is probable that this membrane protein has an aberrant electrophoretic mobility pattern, as seen before for multiple membrane proteins including the ring-forming *c* protein of the $F_1F_0$ ATPase [40].

In an attempt to stabilize the potential interactions between the $X_0$ domain and the $F_1$-like domain, we elected to use a cross-linker to treat the cells before lysis. The reagent Dithiobis(succinimidyl propionate) (DSP) was used, owing to its membrane permeability, its reactivity with primary amines at neutral pH, its relatively short spacer arm (12 Å), and

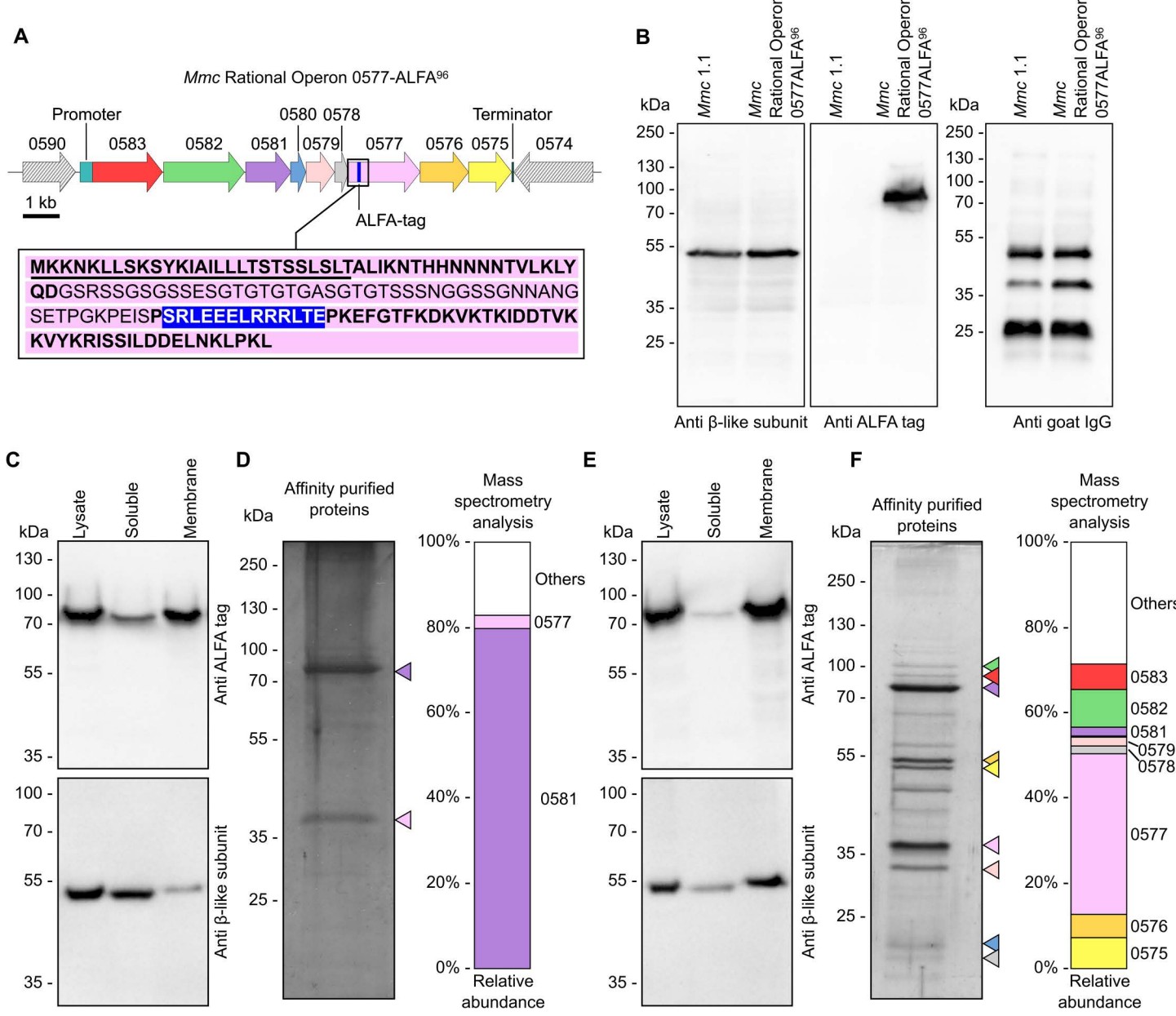

**Fig 4. Purification of MMCAP2_0577-ALFA[96] and the associated F1-like X0 ATPase subunits. A**) Schematic representation of the genomic locus encoding the MIB-MIP-F1-like X0 ATPase subunits in the mutant strain *Mmc* Rational Operon 0577-ALFA[96]. The position of the ALFA tag in MMCAP2_0577 is denoted by a blue bar. A detailed view of the 150 amino acids in N-terminal of MMCAP2_0577-ALFA[96] is given (inset). Underlined characters correspond to the residues predicted to form the signal peptide; characters in light font correspond to the Glycine-Serine rich region; characters in white over blue background correspond to the ALFA-tag. **B**) Western blot analysis of the expression of the β-like subunit, MMCAP2_0577-ALFA[96] tag, and ability to cleave immunoglobulins in the *Mmc* Rational Operon 0577-ALFA[96] mutant strain. **C**) Western blot analysis of the protein content of different cell fractions. Whole cell lysate, and soluble and membrane fractions were separated by SDS-PAGE, then analyzed by Western blotting against either the β-like subunit or the ALFA tag. **D**) Analysis of the co-purification of the putative X0 domain components. After membrane isolation and membrane protein solubilization, affinity chromatography was performed to purify the ALFA-tagged proteins. The purified proteins were separated by SDS-PAGE and stained using colloidal Coomassie blue and silver staining. Colored arrowheads highlight the proteins identified by mass-spectrometry (colors use the same code as in Fig 1). Mass spectrometry was performed to identify the proteins in the sample. The stacked bar chart represents the relative abundance of different proteins of interest. **E**) Western blot analysis of the protein content of different cell fractions after cross-linking of the cells. Whole cell lysate, and soluble and membrane fractions were separated by SDS-PAGE, then analyzed by Western blotting against either the β-like subunit or the ALFA tag. **F**) Co-purification of the putative F1-like X0 ATPase components after cross-linking of the cells. Following membrane isolation

and membrane protein solubilization, affinity chromatography was performed to purify ALFA-tagged proteins. The purified proteins were separated by SDS-PAGE and visualized using Coomassie blue and silver staining. Colored arrowheads highlight the proteins identified by mass-spectrometry (colors use the same code as in Fig 1). Mass spectrometry was performed to identify the proteins in the sample. The stacked bar chart represents the relative abundance of different proteins of interest. *Note: uncropped original images are provided in* S15 Fig.

the possibility to cleave this spacer arm using a reducing agent [41]. In cross-linked condition, fractionation experiments showed that while MMCAP2_0577-ALFA[96] was still located predominantly in the membrane, a larger portion of the β-like subunits were now membrane bound, indicating that the cross-linking was effective (Fig 4E). Purification was subsequently attempted after cross-linking (Fig 4F), and both MMCAP2_0577 and MMCAP2_0581 were again co-purified (bands at ~85 kDa and ~40 kDa respectively). Interestingly, they were accompanied by multiple main bands whose molecular weights were consistent with $F_1$-like proteins. Identification was performed by mass-spectrometry, on four excised pieces of gel covering the mass ranges 250–70 kDa, 70–45 kDa, 45–20 kDa and 20–5 kDa (S5 Table). Overall, proteins of the MIB-MIP system and the $F_1$-like $X_0$ ATPase accounted for over 70% of the abundances in the samples, often being amongst the top five most abundant. Based on apparent mass and intensity, the two closely-spaced bands at ~55 kDa likely correspond to the α-like (MCMAP2_0576)and β-like (MMCAP2_0575)subunits. Similarly, another sharp band at ~30 kDa likely corresponds to the γ-like subunit (MMCAP2_0579). The analysis also reveals the presence of the ε-like subunit (MMCAP2_0578), although no clearly visible band are present for this small (<20 kDa) protein. In addition, we note that while MMCAP2_0580 is found in the mass-spectrometry datasets, the corresponding abundances are systematically low and in ranges that could be considered background noise. This protein is reliably identified in the whole cell mass-spectrometry data, and thus might not be cross-linked to the rest of the complex. Finally, we also observed the presence of two visible bands at ~90–100 kDa, above the main band of MMCAP2_0577. These bands are consistent with the electrophoretic pattern observed for recombinant MIB (MMCAP2_0583) and MIP (MMCAP2_0582), and the presence of both in high abundances is confirmed by the mass spectrometry data.

These biochemical data reinforce the results obtained from the functional mutants, and further solidify the notion that MIB-MIP and the $F_1$-like $X_0$ ATPase are linked and parts of the same processes, as both elements can be co-purified.

## 3. Discussion

Overall, the results presented in this study allow us to attribute a function to Type 3 $F_1$-like $X_0$ ATPases, namely to participate to the evasion from the humoral immune response in mycoplasmas. They act in conjunction with the MIB-MIP system, although the specific mechanism of interactions between the ATPase and MIB and/or MIP remains cryptic and should be the subject of future studies.

Type 3 F-ATPases were initially described in the early 2010's, based on comparative genomics analysis performed on newly available *Mollicutes* genomes [28]. The presence of extra copies of genes related to $F_1F_0$ ATPase subunits in so-called "minimal genomes" was puzzling, especially as these genes were seemingly part of a cluster of 7 genes that did not fully match the content of the 8 genes clusters of the $F_1F_0$ ATPase. Initial phylogenetic classification of these F-ATPases, based exclusively on the α and β subunits homologs led to their segregation in three groups: Type 1 ($F_1F_0$ ATPase), and Type 2 ($G_1$-ATPase) and Type 3 which are more closely related to each other.

While no putative function was at the time available for Type 3, Type 2 ATPase was well known for being involved in the gliding motility of *Mycoplasma mobile* [42]. This model system has been extensively studied over the past two decades, but a full understanding of its mechanism remains elusive [29,42–46]. The gliding machinery of *M. mobile* involves a molecular motor (the Type 2 $G_1$-ATPase), and a set of intracellular (the "jellyfish" structure) and extracellular effectors (the "legs" and "crank arm"). The Type 2 $G_1$-ATPase directly mirrors the Type 3 $F_1$-like $X_0$ ATPase, albeit with a small set of specificities. The Type 2 is encoded by a cluster of 7 genes: MMOB1610 is a membrane protein with 12 transmembrane domains, MMOB1620 is a small cytoplasmic protein, MMOB1630 is a γ-like subunit, MMOB1640

is a ε-like subunit, MMOB1650 is a large protein with an N-terminal signal peptide and two C-terminal transmembrane domains, and MMOB1660 and MMOB1670 are α-like and β-like subunits respectively. Of note, MMOB1650 is much larger than MMCAP2_0577 (~127 kDa compared to ~85 kDa), and has a highly divergent amino-acid sequence resulting in a radically different predicted structure (Uniprotkb: Q6KIC5). Similarly, MMOB1670 is much larger than MMCAP2_0575 (88~kDa compared to ~50 kDa), however this difference is due to the presence of an extra N-terminal domain. The recent structural characterization by electron cryo-microscopy of the Type 2 $G_1$ suggests that this extra domain is involved in the interaction between two $F_1$ hexamers allowing the formation of a so-called "twin motor" [43]. Interestingly, a locus encoding a Type 3 F-ATPase is also found in *M. mobile* and it is at least partially expressed [47]. However, no homologs of MIB or MIP can be identified and thus its role remains cryptic, although it might be a remnant of an ancestral system that lost its functional relevance during evolution.

The current model explaining the role of the Type 2 $G_1$-ATPase in motility proposes that the motor is anchored to both a dedicated cytoskeleton ("jellyfish") and to the membrane through a putative F0 domain. The $F_1$ domain hydrolyses ATP, generating a force and a torque that it then transmits across the membrane to a membrane protein located at the cell surface (the "crank arm") which in turn pulls a substrate binding adhesin (the "leg") (Fig 5A).

We hypothesize that the Type 3 $F_1$-like $X_0$ ATPase might follow a somewhat similar mechanism, with a consumption of ATP by the $F_1$-like domain in the cytoplasm which in turn would energize a conformational change across the membrane, leading eventually to a change in conformation of MIB and/or MIP to force the complex open, release the cleaved antibody and get reset (Fig 5B). We acknowledge that this model is based on analogy to Type 2 $G_1$-ATPases, and that the data gathered to date do not fully support it. Of note, if this model is correct, we would expect to see cleaved immunoglobulins at the cell surface of the ATPase mutants generated in this study. Indeed, the immunoglobulin should still be processed and cleaved properly by MIB-MIP, and thus the MIB-antibody-MIP complex should remain at the cell surface. However, in the cleavage assays presented here, no clear signal corresponding to cleaved IgG or IgM is visible in the cell pellet samples. We propose that this could be due to two factors. First, the number of MIB-MIP systems at the cell surface is likely quite low. In a previous study, we have performed super-resolution imaging of a MIP (MMCAP2_0582) at the surface of *Mmc* cells, and shown that only ~10 individual proteins were present [48]. Thus, surmising that each MIP homolog is present in a similar amount, around ~40 molecules of MIP would be present on a given cell. Given that ~$10^8$ cells are used to generate a sample for Western Blot analysis, we could expect ~$10^9$ cleaved immunoglobulins Heavy Chain

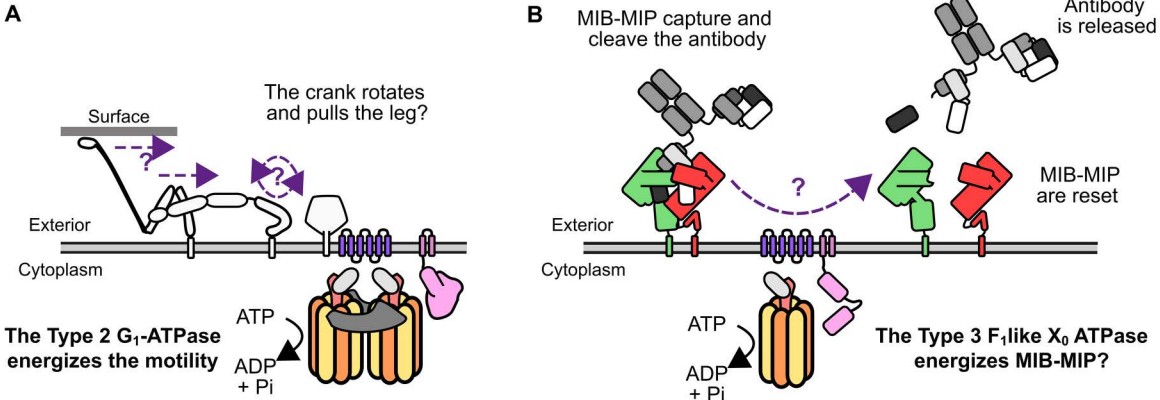

**Fig 5. Model of the function of the Type 2 $G_1$-ATPase and Type 3 $F_1$-like $X_0$ ATPase. A**) Schematic view of the current model of the role of Type 2 $G_1$-ATPase in the motility of *M. mobile*. The ATPase components are colored using the same code as in Fig 1. The other components of the gliding machinery are shown in dark-gray and white. **B**) Schematic view of the proposed model for the role of the Type 3 $F_1$-like $X_0$ ATPase in the MIB-MIP mechanism. The MIB-MIP-ATPase components are colored using the same code as in Fig 1.

which corresponds to ~1 ng of protein. This amount is relatively low, especially when compared to the ~200 ng of cleaved antibody presents in a typical cleavage assay. As a result, cleaved Heavy Chain might be present at the cell surface, but our assay might not be sensitive or specific enough to detect them. A second hypothesis is that our *Mmc* ATPase mutants might present some perturbation of the MIB-MIP system, despite the corresponding locus being intact and the proteins being seemingly present. We do not know yet how the different MIBs, MIPs and the $F_1$-like $X_0$ ATPase are organized in the cell and in particular if there is a permanent or transient co-localization of all the partners. It is unknown if MIB and/or MIP are roaming at the cell surface, randomly searching for antibodies to process then moving to the ATPase, or if they are fixed in place around the ATPase and act as a point-defense system. It is possible that removing the ATPase (*Mmc* ΔATPase), or inhibiting the ATPase from binding and hydrolyzing ATP (*Mmc* 0575-K152A), may have caused a disorganization of the system and thus prevent the proper processing of antibodies.

We also note that the system appears fragile or labile, as cross-linking was required to co-purify the $F_1$-like and $X_0$ domains. In comparison, the typical $F_1F_0$ ATP synthase appears much more stable, with multiple reports in the literature of successful isolation of the full complex from a large number of species [15,49–55]. This stability is promoted by the peripheral stalk, which is comprised of a dimer of b subunits anchored on one side to the membrane via transmembrane segments, and attached on the other side to the δ subunit which caps the $F_1$ assembly. The Type 3 ATPase of mycoplasmas does not present any homolog of the b subunits, although MMCAP2_0577 could potentially play this role owing to its two C-Terminal transmembrane segments. The lability of the $F_1$-like domain could also be related to its function and to a control mechanism. In V-type ATPase, the $V_1$ domain can be disconnected and reconnected to the $V_0$ domain in a highly regulated manner, depending on external factors [56]. In our purification experiments, we observed that a large pool of the $F_1$-like was cytoplasmic, and not associated with the membrane fraction. This cytoplasmic fraction could be uncoupled from the $X_0$ domain, in order to prevent undue consumption of ATP. Indeed, mycoplasmas are known for being ATP-limited, notably because their $F_1F_0$ ATP synthase is understood to function in reverse and to be used only for maintenance of the proton gradient and not for ATP synthesis [57–59]. Thus, from a physiological point of view, it appears important for the cell to be able to regulate this $F_1$-like $X_0$ ATPase in order to strictly limit its hydrolysis of ATP to when it is required for immune evasion.

Interestingly, the MIB-MIP-ATPase system appears to be an evolutionarily complex strategy to counter the host's humoral immune response, as it relies on nine different proteins (MIB, MIP and the seven ATPase subunits) to be functional. In comparison, other bacterial systems dedicated to antibody targeting are usually comprised of a single protein, such as immunoglobulin degrading proteases like the *Ureaplasma* IgA protease [60], the cysteine protease CysP in *Mycoplasma gallisepticum* [16]or the streptococcal IdeS or EndoS [61]; or immunoglobulin binding protein like the staphylococcal Protein A or the streptococcal Protein G [62]. The high degree of conservation of such an intricate system across mycoplasma species is indicative of its efficiency in promoting survival of these pathogens in their host. Analysis of the literature shows that homologs of the MIB-MIP-ATPase system can be found in the proteomics data obtained from multiple species, in axenic conditions or during infection, including *M. mycoides* subsp. *mycoides* [63,64], *Mycoplasma hyopneumoniae* [65], *M. hominis* [66] and *Mycoplasma agalactiae* [67].

It is nonetheless noteworthy that a number of mammalian pathogenic *Mollicutes* have lost part or all of the MIB-MIP-ATPase system. For instance, in the human pathogenic species *M. genitalium* and *M. pneumoniae*, the $F_1$-like $X_0$ ATPase is completely absent, and in place of MIB only Protein M can be found. Protein M appears to be a derivative of MIB, having lost its long N-Terminal tether domain, as well as the Arm domain, maintaining only the Ig-binding MTD domain [68]. Meanwhile, the hemotropic mycoplasmas have all completely lost the system. It is currently unclear which evolutionary forces have driven the losses of the MIB-MIP-ATPase system, but these events indicate that although potent this mechanism is not a panacea for the humoral immune evasion.

The complexity of the MIB-MIP-ATPase system could also be a promising avenue for the development of new strategies to combat mycoplasmas infections. Currently, the options for curative treatment are limited, as only a set of antibiotics

are efficient against mycoplasmas both in the veterinary and medical fields. Furthermore, resistances to these antibiotics are on the rise, threatening to further limit our ability to Fight these diseases [69]. Given the probable importance of the MIB-MIP-ATPase system during the infection, finding a mechanism to block its function could be a way to potentiate the immune response. As the Type 3 $F_1$-like $X_0$ ATPase is highly specific of mycoplasmas and highly divergent from all other known F-type ATPase, it appears to be a potential target for the identification or development of an inhibitor drug [70–72]. Such a drug could be a complement to classical antibiotics, with the added benefit of targeting a non-essential component of the cell potentially limiting the emergence of resistance.

However, further work is required to improve our understanding of this molecular machine. In particular, solving the 3D structure of the ATPase, alone or with MIB and/or MIP will be critical to improve our knowledge and decipher the molecular mechanisms at play.

## 4. Materials and methods

Unless stated otherwise, all experiments were performed at room temperature (20–25°C). Unless specified, the commercial kits were all used according to the manufacturer's recommendations. All sequencing reactions (Sanger, ONT and Illumina) were performed through service providers (Genewiz or Plasmidsaurus).

### 4.1. Bioinformatic analysis

**4.1.1. MIB, MIP and $F_1$-like $X_0$ ATPase homologs identification.** The database Molligen 4.0 (https://services.cbib.u-bordeaux.fr/molligen4) [73] was mined in order to find potential homologs of MIB, MIP and the $F_1$-like $X_0$ ATPase by iterative Blastp through Galaxy's servers (https://usegalaxy.eu/) [74]. To do so, the 224330 CDSs available in the database were collected on 25/11/2019 and pooled in a single file to be used as subject in a Blastp search (https://blast.ncbi.nlm.nih.gov/). The CDS of a representative MIB (MMCAP2_0583), MIP (MMCAP2_0582), and $F_1$-like $X_0$ ATPase (MMCAP2_0581, MMCAP2_0580, MMCAP2_0579, MMCAP2_0578, MMCAP2_0577, MMCAP2_0576 and MMCAP2_0575) were used as query for an initial Blastp search using default parameters and an E-value cut-off of 0.001. A set of 10 hits were randomly selected in the output file and used individually as a new query for a Blastp search against the subject database. This iteration was repeated 10 times. All the hits from the successive result files were then pooled, and duplicates removed, forming the set of putative homologs for each of the representatives.

**4.1.2. Phylogenetic analysis.** Phylogenetic analysis of the homologs of the seven proteins forming the $F_1$-like $X_0$ ATPase were performed using the Phylogeny tool [75] (https://www.phylogeny.fr/) in "One click" mode with default settings and the "gblocks" option disabled. The resulting phylogenetic trees were rendered using iToL [76] (https://itol.embl.de/) and the display options "Circular", "Branch lengths: Ignore", "Branch lengths: Display" and "Branch lengths: Round to 2 decimals".

**4.1.3. Structure prediction using AlphaFold3.** The amino-acid sequences of the proteins of interest were used as input to perform structure predictions using the AlphaFold3 model [77]. For monomer prediction, individual sequences were used. For multimer prediction, a set of multiple sequences were used, with *n* copies of each sequence, with *n* varying depending on the desired predicted complex stoichiometry. Predictions were performed using either ColabFold [78] or Google AlphaFoldServer (https://alphafoldserver.com/). The output was a set of 5 predicted structures and the associated quality metrics (pLDDT: predicted local distance difference test; pTM: predicted template modeling; iPTM: interface predicted template modeling). Structure where visualized using the software ChimeraX [79], and compared using the associated *matchmaker* tool.

### 4.2. Microbial strains and cultivation

All the micro-organism strains used in this study are listed in S6 Table. *Escherichia coli* NEB5alpha (NEB) used for plasmid cloning and propagation were grown at 37°C in LB liquid medium or LB Agar medium, supplemented with the

appropriate antibiotics (kanamycin 50 µg/mL, ampicillin 100 µg/mL, or tetracycline 10 µg/mL). Liquid cultures were performed under agitation in an orbital shaker at 180 rpm. *Mycoplasma mycoides* subsp. *capri* strains (*Mmc*) were cultivated at 37°C in SP5 liquid medium or SP5 Agar medium [21], supplemented with the appropriate antibiotics (tetracycline 5–10 µg/mL or gentamicin 300 µg/mL). All cultures were performed in static condition. *Mycoplasma capricolum* subsp. *capricolum* strain CK ΔRE (*Mcap*ΔRE) used as recipient for transplantation was grown at 30°C in SOB [21] medium without agitation. *Saccharomyces cerevisiae* strains were routinely grown at 30°C in YPDA liquid medium, YPDA Agar solid medium, or SD liquid medium and SD Agar solid medium both supplemented with the appropriate dropout solution (-His, -His-Trp, and -His-Ura-Trp). All liquid cultures were performed under agitation (180–220 rpm).

### 4.3. Molecular biology

Oligonucleotides and plasmids used in this study are listed in S6 Table. PCR primers were purchased from a commercial supplier (Eurogentec). All the PCR reactions performed to amplify DNA cassettes for plasmid cloning or genome editing in yeast used the Q5 DNA Polymerase (NEB). All the PCR reactions performed to screen yeast clones and mycoplasma transplants used the Advantage2 DNA Polymerase Mix (Clontech). All the PCR reactions performed for colony PCR used the Taq DNA Polymerase (NEB). All the multiplex PCR performed to screen yeast clones and mycoplasma transplants used the Multiplex PCR Kit (Qiagen).

#### 4.3.1. Plasmids for the expression of guide RNA in Saccharomyces cerevisiae.
The Cas9 nuclease target sequences were selected by using the "CRISPR Guides" tool available in the Benchling work environment (https://benchling.com). The genome sequence of *Mycoplasma mycoides* subsp. *capri* strain GM12 [80] was used as reference input. Parameters were set to default, except "Design Type: Single Guide," "Guide Length: 20," "Genome: R64–1-1 (sacCer3)," and "PAM: NGG." The target sequence with the highest on-target score in the desired region and the lowest off-target score were selected. The corresponding gRNA expression plasmids were produced by modifying the protospacer sequence of the plasmid p426-SNR52p-gRNA.Y-SUP4t (pgRNA) [34], using the Q5 Site Directed Mutagenesis kit (NEB). After PCR amplification and molecular cloning by Kinase-Ligase-DpnI, the plasmids were transformed by heat shock in chemically competent *E. coli*. The transformed bacteria were plated on LB Agar medium with antibiotic selection and grown overnight. Individual colonies were subsequently screened by colony PCR. Plasmid isolation from a subset of picked clones was performed using the NucleoSpin Plasmid kit (Macherey Nagel). Sanger sequencing was performed to verify the absence of error in the edited locus.

#### 4.3.2. In-yeast recombination templates.
Genome editing in yeast was performed by homologous recombination, which requires a linear DNA fragment comprised (at least) of two sequences homologous to the regions flanking the bacterial genome locus to modify. To construct the *Mmc* ΔATPase mutant strain, the recombination template is comprised solely of two 45-bp long homologous sequences stitched together. The corresponding 90-bp fragment is produced by annealing two complementary 90-base oligonucleotides. To do so, both oligonucleotides were mixed in equimolar amounts in CutSmart buffer (NEB), denatured by heating to 95°C and annealed by gradually cooling to 16°C at the rate of 0.1°/s in a thermocycler.

To construct the mutants *Mmc* 0575-recode and *Mmc* 0575-K152A, the DNA cassette are comprised of edited versions of the entire locus MMCAP2_0576-MMCAP2_0575. To generate these cassettes, the locus MMCAP2_0576-MMCAP2_0575 was amplified from the genome of *Mmc*, and after purification and A-tailing, the PCR products were cloned in pGEMT-Easy (Promega) by ligation. After the molecular cloning reaction, the plasmids were transformed by heat shock in chemically competent *E. coli*. The transformed bacteria were plated on solid LB medium and selected for ampicillin resistance. Individual colonies were subsequently cultivated and used for plasmid isolation using the NucleoSpin Plasmid kit (Macherey Nagel). Plasmids were sequenced to verify the absence of errors. This pGEMT _0576–0575 was then subjected to two successive rounds of mutagenesis, using the Q5 Site-Directed Mutagenesis Kit (NEB), in order to edit the two sites of interest and to generate pGEMT_0576–0575-recoded and pGEMT_0576–0575-recoded+K152A, in

order. After the both edition PCRs, the amplicons were circularized using the Kinase Ligase DpnI mix provided with the kit (NEB). After the molecular cloning reaction, the plasmids were transformed by heat shock in chemically competent *E. coli*. The transformed bacteria were plated on solid LB medium and selected for ampicillin resistance. Individual colonies were subsequently cultivated and used for plasmid isolation using the NucleoSpin Plasmid kit (Macherey Nagel). Plasmids were sequenced to verify the absence of errors. PCR was then performed using the correct plasmids as templates, in order to amplify the 2927 bp recombination cassette, which was then purified using the Illustra GFX PCR purification kit (GE).

To construct the mutant *Mmc* Rational Operon 0577-ALFA[96], the cassette is comprised of two ~290 bp arms flanking the ALFA tag coding sequence. To generate this cassette, the locus MMCAP2_0577 was first amplified from the genome of *Mmc*, and after purification of the amplicon and digestion with BglII, cloned in the plasmid pMycoExp by ligation. After the molecular cloning reaction, the plasmids were transformed by heat shock in chemically competent *E. coli*. The transformed bacteria were plated on solid LB medium and selected for ampicillin resistance. Individual colonies were subsequently cultivated and used for plasmid isolation using the NucleoSpin Plasmid kit (Macherey Nagel). Plasmids were sequenced to verify the absence of errors. This pMycoExp_0577 was then edited, using the Q5 Site-Directed Mutagenesis Kit (NEB), in order to insert the ALFA tag coding sequence at the desired position and yield the plasmid pMycoExp_0577-ALFA[96]. After the edition PCR, the amplicons were circularized using the Kinase Ligase DpnI mix provided with the kit (NEB). After the molecular cloning reaction, the plasmids were transformed by heat shock in chemically competent *E. coli*. The transformed bacteria were plated on solid LB medium and selected for ampicillin resistance. Individual colonies were subsequently cultivated and used for plasmid isolation using the NucleoSpin Plasmid kit (Macherey Nagel). Plasmids were sequenced to verify the absence of errors. PCR was then performed using the correct plasmid as template, in order to amplify the 631 bp recombination cassette.

## 4.4. Generation of *Mycoplasma mycoides* mutant strains

### 4.4.1. Editing of Mmc chromosome cloned in yeast.

The "in-yeast genome edition and back transplantation" method [34] was used to generate the *Mmc* mutant strains. This approach leverages our ability to clone a bacterial chromosome in *S. cerevisiae* and modify it using the genetic tools available in the yeast. The edited bacterial genome can then be isolated and transplanted in a recipient cell, yielding a new mutant bacterial strain. Previously, the genome of *Mmc* GM12 was cloned in the yeast *S. cerevisiae* W303 [33], by the integration of a yeast centromeric plasmid (YCp) in the bacterial genome between the loci MMCAP2_0016 and MMCAP2_0017. This YCp bears a yeast centromere and origin of replication driving the correct replication of the bacterial chromosome, the HIS3 auxotrophic marker to ensure its maintenance, and the *tet(M)* selection marker to select transplanted cells. This YCp-marked bacterial genome can be transplanted to generate the strain *Mmc* 1.1. The yeast *S. cerevisiae* W303–*Mmc* 1.1 was used to generate all the *Mmc* mutant strains in this study. Modification of the *Mmc* 1.1 genome in yeast was carried out using the CRISPR-Cas9 system. First, the yeast W303–*Mmc* 1.1 was transformed with 300 ng of the plasmid p414-TEF1p-Cas9-CYC1t (pCas9), using the lithium acetate method [81]. This plasmid allows the constitutive expression of the *Streptococcus pyogenes* Cas9 nuclease. Yeast transformants were selected and maintained on solid SD-His-Trp medium. The yeast W303–*Mmc* 1.1–pCas9 was then transformed again, using the same lithium acetate protocol, with 300 ng of the plasmid pgRNA and 500 ng of the recombination template. After transformation, the yeasts were maintained 48 hours in liquid SD-His-Trp-Ura medium, then plated on solid SD-His-Trp-Ura medium. Yeast clones were screened to isolate the ones that carried the properly edited bacterial genome. To do so, three successive steps were performed. First, yeast total DNA was extracted and used as a template for a PCR with primers flanking the edited locus. If amplicons of correct size were generated, Sanger sequencing was performed to confirm that they matched the expected design. Alternatively, the PCR products were digested using the appropriate restriction enzyme (NEB) to check for the presence of the edited locus, before Sanger sequencing validation. Subsequently, the total yeast DNA extract was used as a template for a multiplex PCR analysis, using 11 pairs of primers targeting 11 loci of various sizes (ranging from 377 to 1149 bp) spread evenly on

the bacterial genome. This multiplex PCR was used to rapidly assess the integrity of the bacterial genome, and screen out clones in which large genomic regions have been deleted. Finally, the size of the bacterial chromosome carried in the yeast was checked using Pulsed-Field Gel Electrophoresis (PFGE) to screen for potential genomic rearrangements or deletions between two adjacent multiplex PCR loci. To do so, agarose plugs containing the bacterial genomes were prepared [33] using the CHEF Mammalian Genomic DNA Plug Kit (Bio-Rad). Each yeast clone was cultured in 100 ml of SD-His-Trp-Ura medium. The cells were harvested by centrifugation, washed in 50 mM EDTA pH8, and counted on a Malassez counting chamber. Approximately $3 \times 10^8$ cells were then resuspended in Cell Suspension Buffer (Biorad) and embedded in 100 µl of 1% low-melt agarose (Bio-Rad) and cast in a mold to form brick-shaped plugs. After hardening, the plugs were removed from the casts and incubated in a cell lysis solution containing detergents and Proteinase K (Bio-Rad). After a washing step in 20 mM Tris and 50 mM EDTA pH 8 to remove the lysed cell components, the plugs contained only DNA molecules. Before the PFGE, the yeast DNA had to be removed from the plugs. To do so, the plugs were incubated overnight with a cocktail of 30 Units each of the restriction enzymes FseI, RsrII, and AsiSI in 1X Cutsmart buffer (NEB). After restriction, the yeast chromosome fragments were removed from the plugs by standard gel electrophoresis. Under these conditions, the intact circular bacterial chromosomes are not mobile and therefore stay in the plugs. After washing, the plugs were incubated overnight with 30 Units of XhoI in 1X Cutsmart buffer (NEB) to generate three large DNA fragments. The size of these fragments was analyzed by PFGE and compared to that of the expected design.

**4.4.2. Genome transplantation and transplant screening.** In order to yield *Mmc* mutant strains, the edited *Mmc* genomes cloned in yeast were back-transplanted in the recipient cell *Mycoplasma capricolum* strain CK ΔRE [33]. To do so, agarose plugs containing the bacterial chromosomes were digested by incubation with 3 Units β-agarase in 1X β-agarase buffer (NEB). The resulting chromosomal DNA solution was transformed in *Mcap* ΔRE cells using a polyethylene glycol–based protocol [33]. Briefly, the cells from an overnight culture were harvested by centrifugation (5800 g 15 min), washed in Tris 10 mM pH 6.5 NaCl 250 mM, then incubated in $CaCl_2$ 0.1 M on ice for 30 minutes. The cell suspension was then gently overlaid with the chromosomal DNA solution diluted in SP5 ΔFBS medium, mixed with Fusion buffer (Tris 20 mM pH 6.5, NaCl 500 mM, $MgCl_2$ 20 mM, 10% PEG8000) and incubated for 90 minutes at 30°C. PEG contact was stopped by the addition of SP5 medium, followed by cell harvesting by centrifugation (5800 g 15 min). Transformed cells were resuspended in fresh SP5 medium, and selected by plating on solid SP5 Agar medium supplemented with tetracycline. After 4–7 days of incubation at 37°C, individual colonies were collected using a pipette tip to core out a section of solid media, inoculated in liquid SP5 medium supplemented with tetracycline and incubated 24 hours at 37°C. The resulting culture was used to inoculate fresh SP5 medium supplemented with tetracycline, at 1% (v/v) inoculation. This new culture was incubated 24 hours at 37°C. The same passaging process was repeated twice more. At the end of the third passage, a 200 µl sample of the culture was collected and used for transplant screening by PCR. The cells were harvested by centrifugation (6800 g for 10 min), suspended in Tris 10 mM pH 8 EDTA1 mM, and lysed by heating at 95°C for 10 min. The resulting lysate solution was used as template for simplex and multiplex PCR analysis using the same primer pairs as for the yeast transformant screening process. For each mutant strain, a small number of screened clones were then picked and sequenced. Raw sequencing data have been deposited in the SRA under Accession Number PRJNA1359759. To do so, *Mmc* cells grown in SP5 medium were harvested by centrifugation (6800 g 10 min) and used for high molecular weight DNA isolation using the MagAttract HMW DNA kit (Qiagen). Purified DNA quality was checked by agarose gel electrophoresis and spectrophotometry, and quantification was performed using a Qubit 4 fluorimeter and the associated Qubit Broad Range DNA quantification kit (Thermo Scientific). Sequencing was performed by a commercial provider (Plasmidsaurus) using Oxford Nanopore Technology (SQK-RBK114.96 library preparation kit and R10.4.1 flow cells on a PromethION P24 sequencer) to generate long reads, which were assembled (removal of bottom 5% of worst fastq reads via Filtlong v0.2.1 with default parameters; down-sampling the reads to 250 Mb via Filtlong to create an initial sketch of the assembly with Miniasm v0.3; assembly Flye v2.9.1; polishing Flye

assembly via Medaka v1.8.0). This draft assembly was then polished using short reads generated by Illumina sequencing (Nextera XT library preparation) using a custom pipeline run on Galaxy server (removal of Illumina adapters and low-quality reads using trimmomatic in Paired-end mode, with ILLUMINACLIP: Yes - Standard Nextera, SLIDINGWINDOW: 4 bases window - 30 average quality, MINLEN: 100 bases; BWA-MEM2 for paired-end reads in Simple Illumina mode; pilon with default parameters). The polished assembly was compared to the expected sequences using MAUVE, in order to check for genome re-arrangements, Single Nucleotides Polymorphisms and Indels. Validated transplants were stored as cell suspensions in fetal bovine serum at −80°C.

### 4.5. Proteomics

**4.5.1. Sample preparation and protein digestion.** *Mmc* cells grown in SP5 medium were harvested by centrifugation (6800 g 10 minutes), washed in SP5 depleted of fetal bovine serum, and suspended in 1X Laemmli sample buffer with β-mercaptoethanol. After heat denaturation (95°C 10 minutes), the samples were separated by SDS-PAGE on a 10% acrylamide gel. After colloidal blue staining, each lane was excised and further cut into approximately 1 × 1 mm gel pieces. The gel pieces were destained in 25 mM ammonium bicarbonate containing 50% acetonitrile (ACN), rinsed twice with ultrapure water, and dehydrated in ACN for 10 min. After removal of ACN, the gel pieces were air-dried at room temperature, covered with sequencing-grade trypsin solution (10 ng/µL in 50 mM ammonium bicarbonate), rehydrated at 4°C for 10 min, and incubated overnight at 37°C. Peptides were first extracted by incubating the gel pieces for 15 min in 50 mM ammonium bicarbonate with gentle shaking. The supernatant was collected, and extraction was repeated twice with a $H_2O$/ACN/formic acid (47.5:47.5:5, v/v/v) solution for 15 min each. The combined extracts were pooled and dried in a vacuum centrifuge. The resulting peptide digests were resuspended in 0.1% formic acid prior to LC–MS/MS analysis.

**4.5.2. nLC-MS/MS analysis.** Peptide mixtures were analyzed using an Ultimate 3000 nanoLC system (Dionex) coupled to an Orbitrap Fusion Lumos Tribrid Mass Spectrometer (Thermo Fisher Scientific). For total cell extracts, 10 µL of each peptide digest were loaded onto a C18 PepMap trap column (300 µm × 5 mm, LC Packings) at a flow rate of 10 µL/min. Peptides were separated on a C18 PepMap analytical column (75 µm × 50 cm, LC Packings) using a linear gradient of 4–27.5% solvent B over 105 minutes, followed by an increase from 27.5% to 40% solvent B in 10 minutes. Solvent A consisted of 0.1% formic acid in water, while solvent B was 0.1% formic acid in 80% acetonitrile (ACN). The flow rate was maintained at 300 nL/min. For purified fractions, shorter gradients were applied while retaining similar MS parameters.

The mass spectrometer was operated in positive ion mode with a spray voltage of 2 kV. Data were acquired using Xcalibur software in data-dependent acquisition (DDA) mode. Full MS scans (m/z 375–1500) were acquired in the Orbitrap at a resolution of 120,000 (@ m/z 200) with an AGC target of $4 \times 10^5$ and a maximum injection time of 50 ms. Dynamic exclusion was set to 30 s. The most intense precursors were fragmented by higher-energy collisional dissociation (HCD) in top-speed mode within a 3 s cycle.

MS/MS spectra were acquired in the Orbitrap at a resolution of 30,000 with a maximum injection time of 54 ms. Only ions with charge states +2 to +6 were selected for fragmentation. Other parameters were as follows: no sheath or auxiliary gas, capillary temperature 275°C, normalized HCD collision energy 28%, isolation width 1.6 m/z, AGC target $5 \times 10^4$, normalized AGC target 100%, monoisotopic precursor selection (MIPS) set to Peptide, and intensity threshold set to $2.5 \times 10^4$.

**4.5.3. Database search and results processing.** Raw data were processed with Proteome Discoverer 2.5 (Thermo Fisher Scientific) using the SEQUEST HT search engine against a custom database containing all the CDS of *Mmc* GM12 (collected from the Molligen database), excluding spectra for peptides smaller than 350 Da or larger than 5000 Da, while activating the Precursor Detector node. Search parameters included a precursor mass tolerance of 10 ppm and a fragment mass tolerance of 0.02 Da, with only b- and y-ions considered for scoring. Carbamidomethylation of cysteines (+57.021 Da) was set as a fixed modification, while oxidation of methionine (+15.995 Da), methionine loss (−131.040 Da), methionine loss with N-terminal acetylation (−89.030 Da), and protein N-terminal acetylation (+42.011 Da) were specified

as variable modifications, allowing up to two missed trypsin cleavages. Peptide-spectrum matches were validated using the Percolator algorithm, retaining only high-confidence peptides at a 1% false discovery rate (FDR). For strain comparison, label-free quantitation was performed: peak detection and integration used the Minora feature detection node, peptide abundances were normalized to the total bacteria peptide amount, and protein abundance ratios were calculated as the median of all pairwise peptide ratios, with statistical significance assessed via a t-test with Benjamini–Hochberg correction for multiple testing.

### 4.6. Western blot

The primary and secondary antibodies used in this study, as well as the corresponding blocking reagent, dilution factors and incubation durations are listed in S6 Table. Protein samples separated by SDS-PAGE were transferred on a Protran 0.45 nitrocellulose membrane (Cytiva) in a Tris 25 mM glycine 150 mM methanol 10% buffer, using a TE 77 PWR Semi-Dry transfer unit (Cytiva) at the maximum recommended current ($0.8\ mA/cm^2$) for 90 minutes. Membranes were blocked overnight at 4°C in blocking buffer (Phosphate Saline Buffer supplemented with 0.05% v/v Tween 20 and 2% m/v bovine serum albumin), followed by incubation with the primary antibody diluted in blocking buffer (PBS supplemented with 0.05% v/v Tween 20 and blocking reagent) (PBS: NaCl 137 mM; $Na_2HPO_4$ 10 mM; KCl 2.7 mM; $KH_2PO_4$ 1.8 mM; pH 7.5). Unbound primary antibodies were removed by washing the membrane in wash buffer (PBS supplemented with 0.05% v/v Tween 20) three times for 10 min at room temperature. The membrane was then incubated with the secondary antibody (HRP-conjugated) diluted in blocking buffer for 1 h at room temperature. Unbound secondary antibodies were removed by washing the membrane in wash buffer (PBS supplemented with 0.05% v/v Tween 20) three times for 10 min at room temperature. Detection was performed using the Pierce SuperWestPico chemiluminescence substrate (Thermo Scientific) and imaging was performed on a ChemiDoc MP imager (Bio-Rad).

### 4.7. Animal samples

The "infected goat serum" was collected from a goat experimentally infected by *M. mycoides* subsp. *capri* strain 13235. The sample used here was collected 12 days post-infection on the animal B4 [38] and was gifted by Dr. Florence Tardy.

### 4.8. Agglutination assays

*Mmc* cells were grown in SP5 medium until late log phase. 2 mL of culture were sampled and the cells harvested by centrifugation (6800 g 10 min). The supernatant was discarded and the pellet resuspended in 1 mL of fresh SP5 medium with or without supplementation with 2% v/v of infected goat serum B4J12. After 1 hour of incubation, the presence of a precipitate at the bottom of the tube was checked and photographed using an Olympus TG-3 digital camera. Subsequently, the tube was vortexed briefly to resuspend the precipitate, and a 6 µL sample was collected and mounted between a glass slide and a coverslip (Marienfeld). Microscopic observations were performed using a Nikon Eclipse Ti microscope equipped with a Nikon C-DO dark field condenser coupled to a Nikon DS-Qi1Mc camera and a Nikon DS-U3 controller. The samples were further processed by centrifugation (6800 g 10 min). The supernatant was collected, and a 15 µL sample was mixed with ¼ volume of Laemmli 4X loading buffer with reducing agent (β-mercaptoethanol) and denatured by heating at 95°C for 10 minutes. Meanwhile, the cell pellet was washed twice using 500 µL of SP5 without serum and the washed cells collected by centrifugation (6800 g 10 min). After discarding the supernatant, the washed cell pellet was resuspended in 60 µL of PBS. The cells were lysed by 2 cycles of freezing-thawing, after which the viscosity of the sample was reduced by addition of 5 µg of DNAse I from bovine pancreas (Sigma) and incubation for 20 minutes at room temperature. After addition of ¼ volume of Laemmli 4X loading buffer with reducing agent (β-mercaptoethanol), the sample was denatured by heating at 95°C for 10 minutes. Positive controls for immunoglobulin cleavage by MIB-MIP were performed by adding 1 µg each of purified recombinant MIB (MMCAP2_0583) and MIP (MMCAP2_0582) to a 60 µL sample of SP5 medium supplemented with infected goat

serum, and incubating 20 minutes at room temperature before addition of sample buffer and denaturation as above. The presence of IgG and IgM and their potential cleavage in the supernatant and washed cell pellets was checked by analyzing the samples by SDS-PAGE and western-blot.

## 4.9. $F_1$-like $X_0$ ATPase purification

*Mmc* cells were grown in 500 mL of SP5 "NGS" medium, a SP5 medium variant in which the 17% (v/v) of foetal bovine serum are replaced by normal goat serum (Sigma), supplemented with 5 µg/mL of tetracycline. The culture was incubated at 37°C until late log phase (assessed by a pH comprised between 6.3 and 6.8). The cells were harvested by centrifugation (6800 g, 30 min, 15°C), and the supernatant discarded. The cell pellet was then resuspended in 50 mL of PBS, then harvested again by centrifugation (6800 g, 15 min, 15°C). After discarding the supernatant, a second, identical, wash step in PBS was performed. The resulting washed cells pellet was weighed and resuspended in 10 mL of PBS per gram of wet pellet.

When cross-linking was performed, a solution of DSP (ThermoScientific) was added to the washed cells. This stock solution was prepared by dissolving the DSP powder in fresh, anhydrous DMSO (Invitrogen), and added to the cells to yield a final DSP concentration of 1.5 mM. Cross-linking was performed over 30 minutes on a rotating incubator. Cells were then collected by centrifugation (6800 g, 10 min), and the supernatant discarded. The pellet was resuspended in the same initial volume of Lysis Buffer (NaCl 150 mM; Tris 50 mM; $MgCl_2$ 5 mM; Glycerol 20% (v/v); pH 7.5) supplemented with PMSF 1 mM (Sigma), 1 quarter tablet of Complete mini EDTA free protease inhibitor cocktail (Roche) and 2.5 µg/mL of DNAse I (Sigma) and used for cell disruption. The Tris in this buffer was used as quenching reagent to stop the cross-linking reaction. When cross-linking was not performed, the cells were directly resuspended in supplemented Lysis Buffer and disrupted.

After cooling of the suspension in an ice bath, cell disruption was performed by sonication using a VibraCell VCX 500 (Sonics&Materials) and the associated 3 mm probe. Sonication was performed at 30% maximum amplitude, by performing *N* cycles of 8 seconds ON, 59 seconds OFF, while maintaining the sample vessel in an ice bath. The number of cycles was adjusted to reach a total ON time of 1 minute per 1 mL of suspension volume. Lysate clarification was performed by centrifugation (6800 g, 10 min, 4°C) to remove large debris and intact cells. The supernatant was then harvested and used for membrane isolation.

Membrane isolation was performed by ultracentrifugation (100000 g, 90 min, 4°C). After removing the supernatant (cytoplasmic fraction), the membrane pellet was washed by resuspension in one initial volume of Lysis Buffer. The membrane fraction was re-isolated by ultracentrifugation (100000 g, 90 min, 4°C). After removal of the supernatant, the pellet was weighed and either stored at -80°C or used immediately for protein solubilization.

To solubilize the membrane proteins, the membrane pellet was resuspended in 24 mL of Lysis buffer per gram of membrane. The buffer was then supplemented with 1/5th volume of 2.5% DDM (Cube Biotech) in Lysis buffer, yielding a final concentration of 0.5% DDM. Extraction by the detergent was performed over 2 hours at 4°C on a rotating incubator. Solubilized proteins were then separated from the non-soluble fraction by ultracentrifugation (100000 rcf, 90 min, 4°C) and collection of the supernatant.

Purification of the proteins of interest was performed by affinity chromatography of the ALFA tag using the ALFA Selector PE resin (Nanotag). Before use, the resin was equilibrated by washing twice in Lysis buffer supplemented with 0.1% DDM. The equilibrated resin was then added to the solution of solubilized membrane proteins at a ratio of 20 µL of resin per 1.5 mL of solution, and incubated over 2 hours at 4°C on a rotating incubator. Unbound proteins were discarded and the resin washed thrice in 1 mL of Lysis buffer supplemented with 0.1% DDM. Bound proteins were either processed directly for analysis by incubating the beads in Laemmli buffer and denaturing for 10 minutes at 95°C, or eluted by competition. Competition elution was performed by resuspending the resin in Lysis buffer supplemented with 0.5 mM of ALFA peptide (Nanotag) and incubating over 1 hour at room temperature on a rotating incubator. Analysis of the purified

proteins was performed by separating the samples on SDS-PAGE, followed by either Coomassie blue staining or combining Coomassie blue and silver staining of the gel using the ProteoSilver kit (Sigma), performing a Western Blot or by mass spectrometry.

## Supporting information

**S1 Fig. Structure predictions of the $F_1$-like $X_0$ ATPase sub-units.** The 3D structures of the different proteins encoded in the $F_1$-like $X_0$ ATPase locus have been predicted using the AlphaFold3 model. The amino-acid sequence corresponding to the mnemonic MMCAP2_0581-MMCAP2_0575 was used as input. The first model generated for each prediction is presented. The name of each protein is based on the last four characters of the corresponding locus mnemonics. The 3D structures were visualized using ChimeraX and are shown as coloured ribbons (colouring is according to the locus map presented in Fig 1B). When possible, structural comparison to the $F_1F_0$ ATPase sub-units of *E. coli* (PDB: 6OQR) where performed using ChimeraX and the *matchmaker* command. Amino-terminal and Carboxyl-terminal positions are indicated as Nt and Ct, respectively. **A**) Predicted structures of the $F_1$-like sub-unit monomers (top) and comparison to their cognate $F_1$ sub-units (bottom, in grey). **B**) Predicted structure of the $F_1$-like complex generated using AlphaFold3 multimer by inputting multiple copies of several sequences (1x 0579; 1x 0578; 3x 0576; 3x 0575). The predicted complex is presented in two orientations rotated by 90° (left: lateral view; right: axial view). **C**) Predicted structures of the X0 sub-unit monomers. **D**) Predicted structure of the $F_1$-like $X_0$ complex generated using AlphaFold3 multimer by inputting multiple copies of several sequences (1x 0581; 3x 0580; 1x 0577; 1x 0579; 1x 0578; 3x 0576; 3x 0575). The predicted complex structure (left) is presented alongside the experimentally acquired structure of the $F_1F_0$ ATPase of *E. coli* (right, grey, PDB: 6OQR). Scalebar: 10 Å.
(TIF)

**S2 Fig. Phylogenetic analysis of the predicted homologs of MMCAP2_0581.** Phylogenetic analysis of the predicted homologs of MMCAP2_0581 was performed using the Phylogeny tool (https://www.phylogeny.fr/) in "One click" mode. The output phylogenic tree was visualized using iTol (https://itol.embl.de/) as a circular tree. The position of MMCAP2_0581 in the tree is marked by the coloured arrowhead. *Mycoides* cluster and the *Bovis-agalactiae* cluster are highlighted in blue, showing their phylogenetic proximity, in accordance with a horizontal gene transfer scenario, and similarly for the *Hominis* and *Ureaplasma* clusters that are highlighted in green.
(TIF)

**S3 Fig. Phylogenetic analysis of the predicted homologs of MMCAP2_0580.** Phylogenetic analysis of the predicted homologs of MMCAP2_0580 was performed using the Phylogeny tool (https://www.phylogeny.fr/) in "One click" mode. The output phylogenic tree was visualized using iTol (https://itol.embl.de/) as a circular tree. The position of MMCAP2_0580 in the tree is marked by the coloured arrowhead. *Mycoides* cluster and the *Bovis-agalactiae* cluster are highlighted in blue, showing their phylogenetic proximity, in accordance with a horizontal gene transfer scenario, and similarly for the *Hominis* and *Ureaplasma* clusters that are highlighted in green.
(TIF)

**S4 Fig. Phylogenetic analysis of the predicted homologs of MMCAP2_0579 (γ–like subunit).** Phylogenetic analysis of the predicted homologs of MMCAP2_0579 was performed using the Phylogeny tool (https://www.phylogeny.fr/) in "One click" mode. The output phylogenic tree was visualized using iTol (https://itol.embl.de/) as a circular tree. The position of MMCAP2_0579 in the tree is marked by the coloured arrowhead. *Mycoides* cluster and the *Bovis-agalactiae* cluster are highlighted in blue, showing their phylogenetic proximity, in accordance with a horizontal gene transfer scenario, and similarly for the *Hominis* and *Ureaplasma* clusters that are highlighted in green.
(TIF)

**S5 Fig. Phylogenetic analysis of the predicted homologs of MMCAP2_0578 (ε-like subunit).** Phylogenetic analysis of the predicted homologs of MMCAP2_0578 was performed using the Phylogeny tool (https://www.phylogeny.fr/) in "One click" mode. The output phylogenic tree was visualized using iTol (https://itol.embl.de/) as a circular tree. The position of MMCAP2_0578 in the tree is marked by the coloured arrowhead. *Mycoides* cluster and the *Bovis-agalactiae* cluster are highlighted in blue, showing their phylogenetic proximity, in accordance with a horizontal gene transfer scenario, and similarly for the *Hominis* and *Ureaplasma* clusters that are highlighted in green.
(TIF)

**S6 Fig. Phylogenetic analysis of the predicted homologs of MMCAP2_0577.** Phylogenetic analysis of the predicted homologs of MMCAP2_0577 was performed using the Phylogeny tool (https://www.phylogeny.fr/) in "One click" mode. The output phylogenic tree was visualized using iTol (https://itol.embl.de/) as a circular tree. The position of MMCAP2_0577 in the tree is marked by the coloured arrowhead. *Mycoides* cluster and the *Bovis-agalactiae* cluster are highlighted in blue, showing their phylogenetic proximity, in accordance with a horizontal gene transfer scenario, and similarly for the *Hominis* and *Ureaplasma* clusters that are highlighted in green.
(TIF)

**S7 Fig. Phylogenetic analysis of the predicted homologs of MMCAP2_0576 (α-like subunit).** Phylogenetic analysis of the predicted homologs of MMCAP2_0576 was performed using the Phylogeny tool (https://www.phylogeny.fr/) in "One click" mode. The output phylogenic tree was visualized using iTol (https://itol.embl.de/) as a circular tree. The position of MMCAP2_0576 in the tree is marked by the coloured arrowhead. *Mycoides* cluster and the *Bovis-agalactiae* cluster are highlighted in blue, showing their phylogenetic proximity, in accordance with a horizontal gene transfer scenario, and similarly for the *Hominis* and *Ureaplasma* clusters that are highlighted in green.
(TIF)

**S8 Fig. Phylogenetic analysis of the predicted homologs of MMCAP2_0575 (β-like subunit).** Phylogenetic analysis of the predicted homologs of MMCAP2_0575 was performed using the Phylogeny tool (https://www.phylogeny.fr/) in "One click" mode. The output phylogenic tree was visualized using iTol (https://itol.embl.de/) as a circular tree. The position of MMCAP2_0575 in the tree is marked by the coloured arrowhead. *Mycoides* cluster and the *Bovis-agalactiae* cluster are highlighted in blue, showing their phylogenetic proximity, in accordance with a horizontal gene transfer scenario, and similarly for the *Hominis* and *Ureaplasma* clusters that are highlighted in green.
(TIF)

**S9 Fig. Conservation analysis of the α-like sub-unit of the $F_1$-like $X_0$ ATPase.** The amino-acid sequences of 51 representative α-like homologs were extracted from S2 Table, as well as the sequence of the protein AtpA from *E. coli*, and aligned using ClustalW. The alignment quality scores by residue are plotted in purple, and the Walker A motif is highlighted in red (top). The region of the alignment in the black box is detailed below (bottom). The alignment file was opened using JalView (https://www.jalview.org/), and coloured according to the *clustal* palette. The Walker A consensus motif GXXXGKT is noted in grey, aligned with the predicted Walker A motif in the α and α-like subunits. The sequence logo of the Walker A motif was generated using Weblogo (https://weblogo.berkeley.edu).
(TIF)

**S10 Fig. Conservation analysis of the β-like sub-unit of the $F_1$-like $X_0$ ATPase.** The amino-acid sequences of 51 representative β-like homologs were extracted from S2 Table as well as the sequence of the protein AtpD from *E. coli*, and aligned using ClustalW. The alignment quality scores by residue are plotted in purple, and the Walker A motif is highlighted in red (top). The region of the alignment in the black box is detailed below (bottom). The alignment file was opened using JalView (https://www.jalview.org/), and coloured according to the *clustal* palette. The Walker A consensus motif GXXXGKT

is noted in grey, aligned with the predicted Walker A motif in the β and β-like sub-units. The sequence logo of the Walker A motif was generated using Weblogo (https://weblogo.berkeley.edu).
(TIF)

**S11 Fig. Generation of the mutant strain *Mmc* ΔATPase. A**) The locus encoding the MIB-MIP-$F_1$-like $X_0$ ATPase in *Mmc* 1.1 is presented using the same pattern as in Fig 1 (top). The position of the sequence targeted by the guide RNA pgRNA_0577 is denoted by a black diamond. The final locus in the mutant *Mmc* ΔATPase is also displayed (bottom). A detailed view of the recombination arms used to perform the editing in-yeast, following CRISPR-Cas9 stimulated Homologous Recombination is shown (inset). A recombination patch comprised of 2x45 bp homologous arms was generated from two oligonucleotides. The left arm of the patch corresponds to the TAA stop codon of MMCAP2_0582 and the subsequent intergenic spacer. The right arm of the patch corresponds to the terminator of the putative MIB-MIP-$F_1$-like $X_0$ ATPase operon. The double recombination is shown by grey parallelograms. **B**) Simplex PCR screening of the yeast transformants. The properly edited locus should yield a 555 bp amplicon. "1.1": *Mmc* 1.1 gDNA template; "+": positive control; "-": no DNA control. **C**) Multiplex PCR screening of the yeast transformants. The complete genome of *Mmc* mutants should yield the same 11 amplicons as in the positive control. "1.1": *Mmc* 1.1 gDNA template; "WT": *Mmc* GM12 gDNA template; "-": no DNA control. **D**) PFGE analysis of the bacterial chromosome carried in yeast after restriction by XhoI. The complete genome of *Mmc* mutants should yield the same 3 large fragments (590, 269 and 226 kpb) as the positive control. "1.1": *Mmc* 1.1 gDNA. **E**) Simplex PCR screening of the bacterial transplants. The properly edited locus should yield a 555 bp amplicon. The lane 1.1 corresponds to the transplant 1 obtained from the yeast clone 1. The lanes 36.1, 36.2 and 36.3 correspond to the 3 transplants obtained from the yeast clone 36. "WT": *Mmc* GM12 gDNA template; "+": positive control; "-": no DNA control. **F**) Multiplex PCR screening of the bacterial transplants. The complete genome of *Mmc* mutants should yield the same 11 amplicons as in the positive control. "WT": *Mmc* GM12 gDNA template; "-": no DNA control. *Note: only the sample lanes relevant to this publication are annotated. The other samples correspond to other Mmc mutants that are not presented in this publication.*
(TIF)

**S12 Fig. Generation of the mutant strain *Mmc* 0575-recode and *Mmc* 0575-K152A. A**) The locus encoding the MIB-MIP-$F_1$-like $X_0$ ATPase in *Mmc* 1.1 is presented using the same pattern as in Fig 1 (top). The position of the sequence targeted by the guide RNA pgRNA_0575 is denoted by a black diamond. The edited locus in the mutants *Mmc* 0575-recode or *Mmc* 0575-K152A is also displayed (middle and bottom). A detailed view of the recombination arms used to perform the editing in-yeast, following CRISPR-Cas9 stimulated Homologous Recombination is shown (inset). A recombination patch comprised of the complete and edited locus MMCAP2_0576-MMCAP2_0576 was generated by PCR. The left arm of the patch corresponds to the complete wild-type locus MMCAP2_0576. The right arm of the patch corresponds to theedited locus MMCAP2_0575-recode or MMCAP2_0575-K152A. The double recombination is shown by grey parallelograms. **B**) Simplex PCR screening of the yeast transformants. The properly edited locus should yield a 2927 bp amplicon. "+": positive control; "-": no DNA control. **C**) XbaI restriction screening of the Simplex PCR amplicons. The properly edited locus should yield two bands at 1554 bp and 1373 bp. "+": XbaI restriction positive control; "-": XbaI restriction negative control. **D**) Multiplex PCR screening of the yeast transformants. The complete genome of *Mmc* mutants should yield the same 11 amplicons as in the positive control. "1.1": *Mmc* 1.1 gDNA template; "-": no DNA control. **E**) PFGE analysis of the bacterial chromosome carried in yeast after restriction by XhoI. The complete genome of *Mmc* mutants should yield the same 3 large fragments (590, 269 and 226 kpb) as the positive control. "1.1": *Mmc* 1.1 gDNA. **F**) Simplex PCR screening of the bacterial transplants. The properly edited locus should yield a 2927 bp amplicon. The lane 1.1, 1.2 and 1.3 corresponds to the transplants 1, 2 and 3 obtained from the yeast clone 1. "+": positive control; "-": no DNA control. **G**) Multiplex PCR screening of the bacterial transplants. The complete genome of *Mmc* mutants should yield the same 11 amplicons as in

the positive control. "WT": *Mmc* GM12 gDNA template; "-": no DNA control. *Note: only the sample lanes relevant to this publication are annotated. The other samples correspond to other Mmc mutants that are not presented in this publication.*
(TIF)

**S13 Fig. Serum agglutination assays and Immunoglobulin cleavage assays with the mutant *Mmc* strains. A**)
Agglutination of cells by immune goat serum. The cells were grown in axenic conditions, in absence (top) or presence (bottom) of 2% of serum from a goat experimentally immunized, collected 12 days post-inoculation. The bottom of the micro-centrifuge tubes was photographed (top). After resuspension of the flocculates by inversion of the tubes, a sample was mounted between a glass slide and a coverslip and was imaged on a dark-field microscope (bottom - scale bar: 10 μm. **B**) Western blot analysis of the immunoglobulin Heavy Chain cleavage by the MIB-MIP system. Samples corresponding to either culture supernatant or cell pellet derived from the agglutination assays were separated by SDS-PAGE, then analyzed by Western Blotting using primary antibodies targeting either the goat IgG Fc or the goat IgM Heavy Chain. Intact Heavy Chain and MIB-MIP cleaved Heavy Chain are highlighted by a gray and white arrowhead, respectively.
(TIF)

**S14 Fig. Generation of the mutant strain *Mmc* Rational Operon 0577-ALFA[96]. A**) The locus encoding the MIB-MIP-$F_1$-like $X_0$ ATPase in *Mmc* 1.1 is presented using the same pattern as in Fig 1 (top). The locus in the mutant *Mmc* Rational Operon, generated in a previous study, is also shown (middle). The position of the sequence targeted by the guide RNA pgRNA_0577 is denoted by a black diamond. The final locus in the mutant *Mmc* Rational Operon 0577-ALFA[96] is also displayed (bottom). A detailed view of the recombination arms used to perform the edition in-yeast, following CRISPR-Cas9 stimulated Homologous Recombination is shown (inset). A recombination patch comprised of two ~290 bp homologous arms was generated by PCR. The left arm of the patch corresponds to 288 first coding bases of MMCAP2_0577. The right arm of the patch corresponds to the next 298 bp of MMCAP2_0577. The two arms are separated by 45 bp encoding the ALFA tag (13x3 bp) flanked on each side by two serine encoding codons (2x3 bp). The double recombination is shown by grey parallelograms. **B**) Simplex PCR screening of the yeast transformants. The properly edited locus should yield a 1012 bp amplicon. "+": positive control; "-": no DNA control. **C**) Multiplex PCR screening of the yeast transformants. The complete genome of *Mmc* mutants should yield the same 11 amplicons as in the positive control. "1.1": *Mmc* 1.1 gDNA template; "-": no DNA control. **D**) PFGE analysis of the bacterial chromosome carried in yeast. The complete genome of *Mmc* mutants should yield the same 1 large fragment (994 kpb) as the positive control. "1.1": *Mmc* 1.1 gDNA. **E**) Anti-ALFA tag Western blot screening of the bacterial transplants. For each analyzed transplant, whole cell extracts were generated and separated by SDS-PAGE. Anti-ALFA Western blotting was then performed. The *Mmc* mutants should display a single band at ~85 kDa. The lanes 10.1, 10.2, 10.3 and 12.1,12.2, 12.3 corresponds to the 3 transplants obtained from the yeast clone 10 and 12, respectively. "+" recombinant ALFA-tagged protein; "1.1": *Mmc* 1.1 whole cell extract. *Note: only the sample lanes relevant to this publication are annotated. The other samples correspond to other Mmc mutants that are not presented in this publication.*
(TIF)

**S15 Fig. Uncropped original images.** The Figs of this publication contain images that have been cropped and adjusted for contrast or levels. Cropping has been performed to improve legibility or to remove internal control lanes. Adjustments have been applied homogeneously to all images. Image processing does not modify the interpretation of the images. Original uncropped and unadjusted images are provided for reference.
(TIF)

**S1 Table. Mass Spectrometry analysis of the proteome of Mmc GM12.**
(XLSX)

**S2 Table. List of MIB-MIP-ATPase homologs found in the genomes of Mollicutes.**
(XLSX)

**S3 Table. List of the SNPs and INDELS detected in the genomes of *Mmc* mutant strains when compared to the reference genome of *Mmc* 1.1.**
(XLSX)

**S4 Table. Mass Spectrometry analysis of the proteome of *Mmc* 1.1, *Mmc* ΔATPase, *Mmc* 0575-recode and *Mmc* 0575-K152A.**
(XLSX)

**S5 Table. Mass-spectrometry analysis of eluted proteins purified by affinity chromatography using an Anti-ALFA tag resin.**
(XLSX)

**S6 Table. List of the microbial strains, oligonucleotides, plasmids and western blotting antibodies used in this study.**
(XLSX)

## Acknowledgments

The authors would like to thank Dr. Florence Tardy for kindly providing the infected goat serum. The corresponding author would like to thank Leeroy Dusi-Lance for fruitful discussions on the $F_1$-like $X_0$ ATPase. This study was funded by the French National Agency for Research (ANR) grant ANR-21-CE44-0002 ENIgMA.

## Author contributions

**Conceptualization:** Carole Lartigue, Pascal Sirand-Pugnet, Yonathan Arfi.

**Formal analysis:** Julien Berlureau, Laure Bataille, Yonathan Arfi.

**Funding acquisition:** Carole Lartigue, Yonathan Arfi.

**Investigation:** Julien Berlureau, Robin Anger, Emilie Beaulieu, Geraldine Gourgues, Laure Bataille, Jade Jaubert.

**Methodology:** Julien Berlureau.

**Project administration:** Pascal Sirand-Pugnet, Yonathan Arfi.

**Supervision:** Yonathan Arfi.

**Writing – original draft:** Yonathan Arfi.

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
