## [Decision Letter · Decision Letter 0]

27 Jan 2026

PPATHOGENS-D-25-03106

An atypical F-type ATPase is necessary for the function of the antibody cleavage system MIB-MIP in mycoplasmas.

PLOS Pathogens

Dear Dr. Arfi,

Thank you for submitting your manuscript to PLOS Pathogens. After careful consideration, we feel that it has merit but does not fully meet PLOS Pathogens's publication criteria as it currently stands. Therefore, we invite you to submit a revised version of the manuscript that addresses the points raised during the review process.

We look forward to receiving your revised manuscript.

Kind regards,

Mitchell F. Balish, Ph.D.

Academic Editor

PLOS Pathogens

Debra Bessen

Section Editor

PLOS Pathogens

Sumita Bhaduri-McIntosh

Editor-in-Chief

PLOS Pathogens

orcid.org/0000-0003-2946-9497

Michael Malim

Editor-in-Chief

PLOS Pathogens

orcid.org/0000-0002-7699-2064

**Additional Editor Comments:**

The reviews were very positive, with a handful of clarifications, and a consideration from Reviewer #3, that should be addressed in a revised submission.

**Journal Requirements:**

At this stage, the following Authors/Authors require contributions: Julien Berlureau, Robin Anger, Emilie Beaulieu, Geraldine Gourgues, Laure Bataille, Jade Jaubert, Carole Lartigue, Pascal Sirand-Pugnet, and Yonathan Arfi. Please ensure that the full contributions of each author are acknowledged in the "Add/Edit/Remove Authors" section of our submission form.

https://journals.plos.org/plospathogens/s/submission-guidelines#loc-parts-of-a-submission

4) We do not publish any copyright or trademark symbols that usually accompany proprietary names, eg ©,  ®, or TM  (e.g. next to drug or reagent names). Therefore please remove all instances of trademark/copyright symbols throughout the text, including:

- TM on page: 20.

5) Please upload all main figures as separate Figure files in .tif or .eps format. For more information about how to convert and format your figure files please see our guidelines:

6) We have noticed that you have uploaded Supporting Information files, but you have not included a list of legends. Please add a full list of legends for your Supporting Information files after the references list.

7) In the online submission form, you indicated that your data will be submitted to a repository upon acceptance. We strongly recommend all authors deposit their data before acceptance, as the process can be lengthy and hold up publication timelines. Please note that, though access restrictions are acceptable now, your entire minimal dataset will need to be made freely accessible if your manuscript is accepted for publication. This policy applies to all data except where public deposition would breach compliance with the protocol approved by your research ethics board. If you are unable to adhere to our open data policy, please kindly revise your statement to explain your reasoning and we will seek the editor's input on an exemption.

8) Please amend your detailed Financial Disclosure statement. This is published with the article. It must therefore be completed in full sentences and contain the exact wording you wish to be published.

**Reviewers' Comments:**

Reviewer's Responses to Questions

**Part I - Summary**

Reviewer #1: The paper by Berlureau et al., describes a series of elegant mutation experiments to decipher the role of the F1-ATPase (type 3), followed by the application of proteomics (LC-MS/MS) to identify components of the putative F1-ATPase complex together with immunoglobulin cleaving proteins.

Generally the manuscript is well written and clear and the experiments performed with rigor. Having said that, characterizing protein complexes is typically challenging. I am of the opinion that the experiments support the claim that described in the title of the paper “An atypical F-type ATPase is necessary for the function of the antibody cleavage system MIB-MIP in mycoplasmas.” The authors make clear that there are knowledge gaps associated with mechanism and some weaknesses, but these are countered in some regards to the elegant mutational experiments undertaken to address the question understudy.

The authors posit from phylogenetic evidence that horizontal gene transfer is the likely to have occurred several times during the evolution of the Mollicutes. It there evidence for this e.g. G + C content disparity? Is it not possible that the clusters evolved early in the last universal common ancestor and is undergoing decay in specific lineages? – please explain.

Line 228. The authors dispel the detection of many (but not all) mutations because of their presence in intergenic regions within mobile genetic elements. Was there evidence growth rate compliance among all the mutants and wild type- yes you tested those in the 23SrRNA gene but in other cases, like “amino acid substitutions in proteins of unknown function”. The way this is written implies that genes of unknown function are insignificant. There are likely a myriad of important gene functions yet to be determined. Further to this- the section commencing on line 236 indicates that the authors could not find gross differences in proteome content as assessed by western blotting and LC-MS/MS. However these cannot account for any potential posttranslational modification differences

Reviewer #2: This paper found that a protein complex evolved from ATP synthase has a critical role in MIB-MIP system. MIB-MIP system is conserved in Mycoplasma and relatives and protect the bacterial cells from host antibodies by antibody cleavage. The authors conducted several analyses by updated and organized manners, and show evidences supporting their conclusions. Generally, the text is readable but sometimes lengthy. However, readers can select their interested parts.

Reviewer #3: This is an interesting manuscript investigating the role of an atypical type 3 F1 like X0 Mycoplasma ATPase and its role in enabling antibody cleavage by the MIB-MIP system, which this group has also previously thoroughly characterized. The MIB-MIP system is an important and intriguing virulence factor for Mycoplasma species which enables immune evasion and likely plays a massive role in enabling the chronic and persistent nature of Mycoplasma infections. Understanding this system is critical to understanding the virulence of Mycoplasma pathogens such as those belonging to the mycoides cluster, and even other species that have similar, though partially encoded immunoglobulin binding and cleavage systems.

The manuscript is well written and was a joy to read. The authors have done an excellent job in both rigorous experimentation as well as presentation of the data in a way that is easy to read, digest, and understand. The comparative genomics analysis accross 56 different Mollicute species is exceptionally thorough and convincingly demonstrates the conserved co-occurrence between the ATPase gene cluster and the MIB-MIP system. The structural predictions are thoughtfully integrated and provide mechanistic plausibility without overinterpretation. Although some predictions were low confidence, this is a limitation that all in the Mycoplasma field encounter given how diverse Mycoplasma proteins are from the majority of those that are characterized and utilized in the training databases for the structural prediction models. The authors did an excellent job at considering similarities to other ATPases from well modeled species to add credence to the structural predictions.

The experimental work was also technically sophisticated, leveraging the yeast-based genome engineering platform that allowed for the generation of clean and well-validated mutants, including full operon deletion as well as targeted disruption of the Walker A motif that retains expression of the ATPase but prevents its hydrolytic activity. Multiple mutants are generated and verified which really adds to the rigor of the work and collectively points to the direct requirment of ATPase activity for the MIB-MIP system to be functional, which is assessed through agglutination assays in conjunction with assessment of antibody cleavage through western blots. The crosslinking and co-precipitation experiments further indicate a preliminary, but solid biochemical evidence of a physical interaction between MIB-MIP components with the ATPase.

While the methods imparted certain limitations, the authors did a great job at acknowledging these gaps, addressing them in the conclusion, and made sure to not overinterpret the data. This manuscript was a joy to read and review.

**Part II – Major Issues: Key Experiments Required for Acceptance**

Reviewer #1: N/A

Reviewer #2: L103) Description and structure prediction of the putative F1-like X0 ATPase of Mmc:

I recommend to include a typical predicted structure like Fig. S1B or D in Fig. 1.

L321) We eventually succeeded by adding the sequence of the ALFA-tag, a 13 amino-acids epitope tag for which a cognate Nanobody is available [39], after the residue S96 (Fig 4A).

The authors should show evidences of “membrane” fraction, based on mass spec, electron microscopy, optical microscopy and so on.

L422) assay. As a result, cleaved Heavy Chain might be present at the cell surface, but our assay might not be sensitive or specific enough to detect them.

How about possibility to track antibodies with an intense fluorescent dye as a future work?

Reviewer #3: I did not identify any major issues requiring further experimentation that is required for acceptance.

**Part III – Minor Issues: Editorial and Data Presentation Modifications**

Reviewer #1: Minor considerations:

The following needs to be addressed.

1. Line 252 correspond not corresponds

2. Line 327. Please explain “a detergent selected after an initial screening (data not shown). What type of screening are you meaning- please explain the statement.

3. Line 463. The proteomic determination of the F1ATPase components were first identified in M. hyopneumoniae in the paper by Tacchi et al., 2016 (Open Biol. 2016 Feb;6(2):150210. doi: 10.1098/rsob.150210.-supplementary data), preceding Reference 65.

Reviewer #2: L62) by twisting the variable domains of the Heavy and Light Chains of immunoglobulins.

Explain the information to suggest twisting.

L90) and the rare Type 2 which corresponds to the molecular motor involved in energizing the motility in Mycoplasma mobile [29,30].

The ATPase unit of Type 2 ATPase is designated also as “G1-ATPase” in reference [43]. Refer to this name at an early position.

L129) and many) F1-like complex

Check subscript, italic, capitarization, especially for “References”.

L191) an extremely strong genetic linkage

This sounds too strong. “strong genetic linkage” may be enough.

L271) exactly the same agglutinated phenotype,

This sounds too strong. The pellet appearances are not the same. “very similar agglutination phenotype” may be enough.

L293) 0575-recode

Does this mean a 0575 mutant complemented by an original 0575? Clarify it.

L302) We eventually succeeded by adding the sequence of the ALFA-tag, a 13 amino-acids epitope tag for which a cognate Nanobody is available [39], after the residue S96 (Fig 4A).

Which protein did you choose? Clarify it.

L405) which in turn pulls on a substrate binding adhesin (the “leg”) (Fig 5A).

Is “on” necessary?

Reviewer #3: Not necessarily an issue, but I did wonder about whether the authors looked into whether binding of antibody to the cognate antigen on Mmc is required for the activation of the MIB-MIP system, or if the system is able to cleave non-specific antibodies that the system actively binds out of the supernatant/medium in which it exists in? Does the MIB-MIP system bind and cleave non-specific antibodies?

Could it be that the ATPase is serving more so to localize the MIB-MIP system to the site of antibody bound to the surface of the cell rather than energizing the MIP to cleave? Or is the current thought that MIB just binds whichever antibody is in close proximity to it? Just a couple of comments meant out of curiosity.

PLOS authors have the option to publish the peer review history of their article (what does this mean? ). If published, this will include your full peer review and any attached files.

**Do you want your identity to be public for this peer review?** For information about this choice, including consent withdrawal, please see our Privacy Policy .

Reviewer #1: **Yes:** Steven P. Djordjevic

Reviewer #2: **Yes:** Makoto Miyata

Reviewer #3: No

**Figure resubmission:**
---

## [Editor Report · Decision Letter 1]

28 Feb 2026

Dear Dr Arfi,

We are pleased to inform you that your manuscript 'An atypical F-type ATPase is necessary for the function of the antibody cleavage system MIB-MIP in mycoplasmas.' has been provisionally accepted for publication in PLOS Pathogens.

Best regards,

Mitchell F. Balish, Ph.D.

Academic Editor

PLOS Pathogens

Debra Bessen

Section Editor

PLOS Pathogens

Sumita Bhaduri-McIntosh

Editor-in-Chief

PLOS Pathogens

orcid.org/0000-0003-2946-9497

Michael Malim

Editor-in-Chief

PLOS Pathogens

orcid.org/0000-0002-7699-2064

All issues that needed addressing have been adequately addressed.
---

## [Editor Report · Acceptance letter]

Dear Dr Arfi,

We are delighted to inform you that your manuscript, "An atypical F-type ATPase is necessary for the function of the antibody cleavage system MIB-MIP in mycoplasmas.," has been formally accepted for publication in PLOS Pathogens.

Best regards,

Sumita Bhaduri-McIntosh

Editor-in-Chief

PLOS Pathogens

orcid.org/0000-0003-2946-9497

Michael Malim

Editor-in-Chief

PLOS Pathogens

orcid.org/0000-0002-7699-2064